# The Impact of Calcium Overload on Cellular Processes: Exploring Calcicoptosis and Its Therapeutic Potential in Cancer

**DOI:** 10.3390/ijms252413727

**Published:** 2024-12-23

**Authors:** Adrianna Gielecińska, Mateusz Kciuk, Renata Kontek

**Affiliations:** 1Department of Molecular Biotechnology and Genetics, University of Lodz, Banacha 12/16, 90-237 Lodz, Poland; adrianna.gielecinska@edu.uni.lodz.pl (A.G.); mateusz.kciuk@biol.uni.lodz.pl (M.K.); 2Doctoral School of Exact and Natural Sciences, University of Lodz, Matejki Street 21/23, 90-237 Lodz, Poland

**Keywords:** calcium, cancer, apoptosis, necroptosis, necrosis, calcicoptosis

## Abstract

The key role of calcium in various physiological and pathological processes includes its involvement in various forms of regulated cell death (RCD). The concept of ‘calcicoptosis’ has been introduced as a calcium-induced phenomenon associated with oxidative stress and cellular damage. However, its definition remains controversial within the research community, with some considering it a general form of calcium overload stress, while others view it as a tumor-specific calcium-induced cell death. This review examines ‘calcicoptosis’ in the context of established RCD mechanisms such as apoptosis, necroptosis, ferroptosis, and others. It further analyzes the intricate relationship between calcium dysregulation and oxidative stress, emphasizing that while calcium overload often triggers cell death, it may not represent an entirely new type of RCD but rather an extension of known pathways. The purpose of this paper is to discuss the implications of this perspective for cancer therapy focusing on calcium-based nanoparticles. By investigating the connections between calcium dynamics and cell death pathways, this review contributes to the advancement of our understanding of calcicoptosis and its possible therapeutic uses.

## 1. Introduction

Homeostasis is a state of dynamic equilibrium within the body, where various physiological parameters are regulated and maintained within a narrow range of accepted standards. This balance is essential for ensuring the stability and proper functioning of all organs and systems, allowing the body to respond effectively to internal and external changes while sustaining overall health and survival. It can, however, be considered at the level of individual processes, systems, or cell types. Accordingly, oxygen homeostasis maintains proper oxygen levels, preventing hypoxia and oxidative stress. Immune cell homeostasis ensures effective protection against infections while preventing autoimmune diseases. Neurotransmitter homeostasis ensures the proper functioning of the nervous system, preventing nerve dysfunction and neurological and mental disorders. Maintaining stable internal conditions despite a changing external environment is essential for optimal functioning. The disruption of key homeostasis components can lead to pathological conditions and disease development. Calcium is one of the most important elements in the body and plays a key role in many physiological processes, making its homeostasis extremely important.

Calcium serves both structural and functional roles in the body. Structurally, it is a critical component of bone and teeth, providing strength and rigidity. Functionally, calcium is essential in a variety of physiological processes, including muscle contraction of skeletal, cardiac, and smooth muscles. In this process, calcium ions enter muscle cells and initiate the interaction between actin and myosin filaments, leading to contraction [1,2]. It is also essential for the proper depolarization and repolarization of heart muscle cells, ensuring a proper cardiac rhythm [3]. Calcium influences neurotransmitter release, neuronal membrane stabilization, and synaptic transmission, which are fundamental for nerve conduction and neuron communication. Additionally, it acts as a cofactor for enzymes in blood clotting and regulates the secretion of hormones like insulin from the pancreas [4]. Disturbances in calcium homeostasis can lead to hypocalcemia or hypercalcemia, causing conditions such as bone demineralization, cardiac arrhythmias, hypertension, convulsions, muscle weakness, nephrolithiasis, inflammation, and diseases like osteoporosis, tetany, metabolic disorders, cardiovascular diseases, and cancer [5,6]. Recently, it was reported that calcium can also induce cell death through a phenomenon defined as calcicoptosis [7].

While some scientists consider calcicoptosis a form of regulated cell death (RCD), its exact nature and classification remain topics of ongoing discussion. This uncertainty stems from differing viewpoints, with some researchers interpreting calcicoptosis as a general response to calcium overload, while others argue it may specifically refer to a tumor cell death mechanism induced by calcium ion accumulation [8,9]. In 2019, this debate was intensified by the work of Zhang et al., who introduced sodium hyaluronate-modified calcium peroxide-loaded nanoparticles designed to degrade in the acidic tumor microenvironment, releasing free calcium ions [7]. These ions accumulated, resulting in tumor calcification and the death of cancer cells through disruption of signaling pathways and increased oxidative stress. The discovery of calcium-induced cell death and the development of targeted carriers for cancer cells have generated considerable interest in this form of cell death that could open new therapeutic avenues for cancer treatment. Moreover, there have been numerous reports linking calcium overload to apoptosis, necroptosis, pyroptosis, and other forms of regulated cell death (RCD). Each of these processes is characterized by distinct molecular mechanisms which we have described in more detail in our previous articles [10,11]. Calcium ion overload, introduced as a mechanism leading to calcicoptosis, prompts further exploration into whether it represents a distinct form of cell death or operates within already known pathways.

In this article, we will discuss the role of calcium in key signaling and metabolic processes, summarize previous reports on calcicoptosis, and describe the impact of calcium ion overload on various forms of cell death.

One of the primary aims of this review is to investigate whether calcicoptosis represents a distinct, independent type of RCD or if calcium overload is a phenomenon observed across multiple RCD pathways. We will analyze the molecular pathways and markers associated with calcium overload and its potential involvement in different forms of cell death. Our goal is to provide a comprehensive understanding of calcicoptosis and offer diverse perspectives on its functional mechanisms, particularly in the context of cancer treatment.

## 2. The Physiological and Pathological Role of Calcium in the Human Body

Calcium is a chemical element of the group of alkaline earth metals. In the body, it occurs mainly in the form of calcium ions, the level of which is controlled by homeostatic mechanisms. More than 99% of calcium is stored in bone tissue, the rest in plasma and body fluids. Typically, the level of total calcium in the blood serum is 9 to 10.5 mg/dL, which corresponds to approximately 2.25 to 2.62 mmol/L [12,13]. Although it is not the most abundant element in the human body, it plays a fundamental role in many important physiological processes, including regulation of synaptic transmission, muscle contraction, heart rate, blood clotting, structural integrity of bones and teeth, control of hormone secretion, and other metabolic processes in the body.

To precisely control intracellular calcium levels, cells have evolved sophisticated mechanisms to regulate calcium homeostasis, meticulously modulating its movement into and out of cells and organelles. This system includes various types of pumps, channels, and exchangers located in the endoplasmic reticulum (ER) and cell membrane, which maintain a calcium concentration gradient in different cellular compartments. The ER is one of the most critical calcium stores within the cell [14]. The release of calcium ions from the ER via inositol triphosphate receptors (IP3Rs) and ryanodine receptors (RyRs) increases cytoplasmic calcium levels, activating a cascade of calcium-dependent events in response to stimuli [15,16]. Sarcoplasmic/endoplasmic reticulum Ca^2+^-ATPase (SERCA) pumps work against the concentration gradient to fill the ER with calcium ions [17]. In contrast, the calcium release-activated calcium channel protein 1 (ORAI1) and the ER stromal interaction molecule 1 (STIM1) proteins present in the cell membrane cause the entry of ions from outside the cell [18]. Changes in the expression of these proteins may be associated with the development of various diseases, including cancer, or impact cancer drug resistance and metastasis.

One of the most important functions of calcium in the body is its participation in synaptic transmission. Calcium plays a crucial role in the release of neurotransmitters, the modulation of postsynaptic responses, and the regulation of synaptic plasticity. This process begins with an action potential that propagates along the axon of the presynaptic neuron and reaches the synaptic terminal. Depolarization of the cell membrane at the synaptic terminal causes the opening of voltage-dependent calcium channels of the L type (Cav1), N type (Cav2.2), and P/Q type (Cav2.1) in the presynaptic membrane [19]. Their opening allows calcium ions to flow into the presynaptic cell from the extracellular space. This influx of calcium ions is crucial for the exocytosis of synaptic vesicles. The increase in calcium concentration is detected by proteins such as synaptotagmin, which are bound to the membrane of synaptic vesicles. After binding to calcium ions, synaptotagmin initiates interactions with other proteins in the synaptobrevin, syntaxin, and SNAP-25 complex (SNARE), leading to the fusion of the vesicle with the presynaptic membrane and release of neurotransmitters into the synaptic cleft [20]. These neurotransmitters then bind to receptors on the postsynaptic membrane, causing the channels to open or close, leading to depolarization (excitatory postsynaptic potential) or hyperpolarization (inhibitory postsynaptic potential) of the postsynaptic neuron [21]. Depending on the type of receptors and ions involved, this effect may excite or inhibit the activity of the postsynaptic neuron. After neurotransmission, excess calcium ions must be removed from the presynaptic terminal to restore the cell to its resting state. Calcium is removed by calcium pumps (calcium ATPases such as SERCA2) and sodium–calcium exchangers, which transport it back to the extracellular space or the endoplasmic reticulum [22]. Calcium levels are therefore tightly controlled by various feedback mechanisms and transporter proteins. Calcium also plays a significant role in synaptic plasticity, which is the basis of learning and memory processes. Synaptic plasticity refers to the ability of synapses to change the strength of their connections in response to neuronal activity. The main mechanism controlling synaptic strength is the modulation of α-amino-3-hydroxy-5-methyl-4-isoxazolepropionic acid (AMPA) receptors in the postsynaptic membrane. Both long-term potentiation and long-term depression are dependent on increases in calcium concentration in the postsynaptic neuron [23,24]. An increase in calcium concentration initiates a signaling cascade that leads to the activation of several proteins, mainly kinases, which, by phosphorylation, activate AMPA receptors and other synaptic proteins, consequently strengthening the synapse [25]. In the case of long-term depression, where there is a smaller but prolonged increase in calcium levels, protein phosphatases are activated, causing dephosphorylation of AMPA receptors, resulting in a weakening of synaptic strength. Dysfunctions in the regulation of calcium levels may lead to impaired synaptic plasticity, contributing to the development of neurological diseases [26].

Disturbance of calcium contributes significantly to neurodegenerative diseases and neuronal damage. Compounds such as ReS19-T have been shown to restore calcium balance in Alzheimer’s disease models by inhibiting pathological calcium channel activation, reducing calcium influx, and improving neuronal function [27]. Similarly, calcium dysregulation plays a key role in neuronal death after cerebral ischemia, where impaired calcium regulation triggers ER stress and mitochondrial dysfunction, leading to cell death [28]. Targeting ER stress pathways and proteins regulating intracellular calcium offers promising therapeutic strategies for both neurodegenerative disorders and ischemic brain damage [27,28].

Calcium is a fundamental mineral that, in combination with phosphate, creates hydroxyapatite, which is a scaffold for connective tissue, ensuring mechanical strength and resistance to abrasion of bones and tooth enamel. In calcification (mineralization), hydroxyapatite is deposited on collagen fibrils, making the bone tissue both flexible and hard [29]. The minerals contained in it undergo constant changes during bone formation, resorption, and remodeling, i.e., removal of old bone tissue and creation of new bone tissue. These processes are precisely regulated by thyroid and parathyroid hormones, calcitriol (1,25-hydroxyvitamin D3), prostaglandins, and other factors [30]. Hydroxyapatite can store and exchange calcium ions, which is important in maintaining mineral homeostasis in the body. Changes in the structure of hydroxyapatite may lead to various abnormalities of the skeletal system, such as osteoporosis or tooth decay [29]. Another threat is pathological calcification, which occurs in soft tissues and is caused by the accumulation of calcium phosphate crystals, also called calcified deposits. Their presence may lead to changes in mechanical properties, tissue stiffness, and inflammation [30]. In extreme cases, calcification may lead to phenotypic changes in tissues that become bone-like. Calcifications occurring in tissues are therefore the result of abnormalities in regulatory mechanisms that actively prevent this process.

The relationship between cell death and calcification was investigated in human vascular smooth muscle cells [31]. Apoptotic bodies, which are normally cleared by phagocytosis, can initiate calcification in diseased blood vessels under stress conditions like oxidative stress and modified lipids [30,32]. Impaired clearance of apoptotic bodies fosters secondary necrosis, creating an environment for calcium deposition. Calcium phosphate crystals exacerbate this process by increasing cytosolic calcium levels and causing membrane damage, leading to apoptosis or necrosis [31]. Strategies like inhibiting lysosomal activity or using calcification inhibitors like fetuin-A can mitigate crystal-induced cell death. However, in diseases like chronic kidney disease, low fetuin-A levels promote severe vascular calcification due to increased calcium deposition [31].

Calcium ions act as transmitters in muscle contraction in both striated (skeletal and cardiac) and smooth muscles, initiating and regulating interactions between contractile proteins. In striated muscle, the process starts with a nerve impulse that depolarizes the muscle cell membrane (sarcolemma) and travels through the T-tubules [33]. This leads to the opening of calcium channels in the sarcoplasmic reticulum, releasing calcium into the sarcoplasm. Calcium then binds to troponin, causing a conformational change in the troponin–tropomyosin complex, which exposes the binding sites to actin [34]. The myosin heads form cross-bridges with these sites, and adenosine triphosphate (ATP) hydrolysis provides the energy for muscle contraction. For muscle relaxation, calcium is pumped back into the sarcoplasmic reticulum, causing tropomyosin to return to its original position, blocking the actin–myosin interaction. Troponin is absent in smooth muscle and contraction is controlled by calcium-binding calmodulin, which activates myosin light chain kinase (MLCK). MLCK phosphorylates myosin regulatory light chains (RLCs), enabling the interaction of myosin with actin and determining the contraction process. However, a decrease in calcium ions causes RLC dephosphorylation by MLC phosphatases and muscle relaxation. The work of these two enzymes causes the muscle contraction and relaxation process to proceed slower than in the case of striated muscles [34]. Each muscle type has specific calcium channels and proteins for calcium homeostasis. Calcium release in skeletal muscles is initiated by the depolarization of the muscle fiber cell membrane, which leads to the activation of dihydropyridine receptors (DHPRs), acting as voltage sensors in the T-tubule membrane. Their conformational change regulates the opening of RyRs, allowing calcium ions to flow out from the sarcoplasmic reticulum [35,36]. Cardiac muscles have the same channels and additional ones for electromechanical coupling, while smooth muscles utilize receptor-associated calcium channels and mechanisms independent of the sarcoplasmic reticulum. The regulation of calcium levels and the diversity of proteins that control these processes emphasize the complexity of cellular mechanisms adapted to the specific structure and function of different types of muscles.

Rhabdomyolysis involves the breakdown of skeletal muscle fibers, releasing their contents into the bloodstream and causing complications like kidney failure, electrolyte imbalances, and organ damage [37]. Mechanical injury, drug poisoning, infections, or metabolic disorders can cause it. The resulting loss of electrolyte homeostasis triggers calcium influx, muscle contraction, ATP depletion, and membrane damage [38]. Excess calcium activates enzymes that degrade myofibrils and membrane proteins, releasing myoglobin, which can clog renal tubules. Calcium influx also exacerbates cell death through apoptosis and necrosis, while blocking calcium channels reduces muscle toxicity [39].

The examples presented above are only the main aspects of the physiological role of calcium in the body, focusing on its essential participation in fundamental life processes of cells and the entire organism. Calcium homeostasis is precisely controlled by various mechanisms, abnormalities of which can contribute to disease development and even mortality. However, a detailed exploration of calcium’s functions and regulatory mechanisms is not the aim of this review. Our intention is only to emphasize the critical importance of calcium in maintaining proper bodily function.

Calcium signaling plays a critical role in cancer progression by modulating cellular behaviors essential for tumor growth and metastasis. Dysregulated calcium homeostasis in cancer cells promotes increased proliferation, survival, and migration—hallmarks of cancer [40,41,42]. Alterations in calcium channels and pumps, particularly in store-operated calcium entry (SOCE) pathways, enhance calcium influx, which drives cancer cell invasion [43]. Studies show that calcium channels like transient receptor potential melastatin 7 (TRPM7) and transient receptor potential cation channel subfamily V member 4 (TRPV4) are key players in supporting endothelial cell migration, contributing to angiogenesis, which is crucial for tumor growth [44,45]. Furthermore, the remodeling of calcium signaling affects the epithelial-to-mesenchymal transition, a process that facilitates cancer cell dissemination and metastasis [46]. Differences in calcium signaling between tumor and non-tumor cells have been identified in various cancer models, such as in bladder, breast, and melanoma cells, where changes in calcium influx and oscillation patterns are linked to increased migratory and invasive potential [47]. Understanding these alterations in calcium signaling provides opportunities for therapeutic interventions aimed at disrupting tumor progression, highlighting how selective targeting of calcium-related pathways could limit cancer cell survival and metastasis while sparing normal tissue. The following section will delve deeper into the role of calcium signaling in cell death, discussing the underlying mechanisms and potential strategies for inducing tumor cell apoptosis or necrosis.

## 3. Calcicoptosis as a New Form of Regulated Cell Death

Cell calcium overload has long been associated with disruption of metabolic and proliferative processes [48,49,50,51]. As previously mentioned, it is caused by the excessive influx of calcium ions across the cell membrane via ion channels or by the release of calcium from intracellular stores such as the ER and mitochondria. Long-term dysfunction in calcium regulation leads to cell overload. An increase in intracellular calcium concentration is often accompanied by oxidative stress, which causes many destructive effects, ultimately leading to cellular damage. In properly functioning cells, innate cellular mechanisms will counteract the overload of calcium ions. However, under conditions of oxidative stress, their regulatory capacity decreases. This synergistic effect was observed in mitochondria, whose structure and function were altered, resulting in apoptotic cell death [52,53].

The concept of ‘calcicoptosis’ as a new type of cell death was recently proposed by Zhang et al., who developed calcium peroxide nanoparticles (SH-CaO_2_ NPs) coated with sodium hyaluronate [7]. These nanoparticles were stable in body fluids, but under the influence of the acidic environment of the tumor, they caused the release of calcium ions and hydrogen peroxide. Decreased levels of catalase in cancer cells promoted the development of oxidative stress, leading to changes in calcium channels such as desensitization and the retention of calcium ions in cells [7]. Calcium overload causes irreversible mitochondrial damage and other cellular pathologies. In the 4T1 subcutaneous model, researchers observed significant reductions in tumor size and induction of cancer cell death. However, in orthotopic models, inhibition of tumor growth was not accompanied by metastases, confirming the therapeutic effectiveness of the designed nanoparticles while minimizing side effects [7]. Attention was also paid to the potential of cell calcification as an additional tumor-suppressing mechanism [7].

### 3.1. Calcium Overload in Apoptosis

Treatment with chemotherapy and radiation generates large amounts of free radicals in cancer cells, which may promote the accumulation of calcium ions and the process of tissue calcification. Excess calcium and free radicals fuel each other and enhance their anticancer effects, resulting in cancer cell death. The detailed mechanism of this process remains unknown. Nevertheless, calcium signaling regulates various cellular processes, including metabolism, proliferation, and cell death. Calcium ion overload has been observed to induce various forms of cell death, such as apoptosis, necrosis, necroptosis, pyroptosis, and autophagy [54,55,56,57,58,59,60,61,62].

Apoptosis is a fundamental type of RCD that plays a key role in maintaining homeostasis and eliminating damaged or dangerous cells. We discussed the detailed mechanism of apoptosis in our previous article [10]. Under the influence of various stressors, such as ER stress caused by the accumulation of misfolded proteins, DNA damage, oxidative stress, or pro-apoptotic signals, calcium is released into the cytoplasm through IP3R and RyR channels located in the ER membrane. A high concentration of calcium in the cytoplasm activates calpains, which are calcium-dependent cysteine proteases [63,64]. Calpains can affect caspase 12 enzymes, leading to their activation. Caspase 12 is an initiator protein that triggers a cascade of apoptotic signals, activating executioner caspases. This process is known as the ER-associated apoptosis pathway due to its close association with ER and calcium ion imbalance [64]. Calpains may also influence pro-apoptotic proteins like Bcl-2 associated X protein (BAX) and BH3 interacting-domain death agonist (BID), which, by forming pores in the mitochondrial outer membrane, facilitate the release of cytochrome C from mitochondria and initiate the intrinsic apoptotic pathway [64].

Microdomains may form near calcium channels, which are concentrated areas with high calcium levels resulting from the opening of these channels in the cell membrane and the subsequent influx of calcium. These microdomains can regulate gene transcription pathways due to their precisely localized and high ion concentrations, allowing them to induce specific cellular reactions [65]. Calmodulin is highly sensitive to changes in calcium concentration, and its binding to calcium causes a conformational change. Activated calmodulin can regulate the functioning of proteins, such as protein phosphatases, which affects Bcl2-associated agonist of cell death (BAD) by causing its translocation to mitochondria and neutralizing the action of anti-apoptotic proteins [65,66]. In addition, it can be transported to the nucleus via the γ subunit of calcium/calmodulin-stimulated protein kinase II (CaMKIIγ), where it participates in the phosphorylation of transcription factors such as the cAMP-responsive element binding protein (CREB) [65]. CREB regulates the transcription of genes responsible for neuroplasticity, memory, cell development and differentiation, response to oxidative stress, and cell survival [67]. Nuclear factor of activated T cells 1 (NFAT1) is a transcription factor that regulates gene expression in response to calcium signals. NFAT1 is localized in the cytosol, and its activity is inhibited by phosphorylation. An increase in calcium levels caused by the opening of calcium channels such as ORAI1 may lead to the dephosphorylation of NFAT1 by calcineurin. Dephosphorylated NFAT1 is then translocated to the cell nucleus, where it activates specific transcription programs, potentially resulting in increased expression of genes controlling proliferation, immune response, or other physiological processes. Calcium signals can vary in amplitude, spatial, and temporal characteristics [65]. Long-term increases in calcium can induce apoptosis, while increases in mitochondrial calcium can affect ATP production levels. Changes in cytosolic calcium levels may regulate cell migration by affecting the regulation of actin and cytoskeletal structure, as well as the activity of adhesion proteins and complexes involved in the formation and breakdown of cell adhesions. The specificity of the cell’s response to calcium signals depends on the precise location of the microdomains and the types of kinases and transcription factors involved. In the case of NFAT4, calcium increases are required in both the microdomains and the nucleus, highlighting the context specificity of calcium signals [65].

The SOCE process is one of the most important mechanisms regulating the opening of calcium channels and the influx of calcium into the cell from the ER and the extracellular environment. The STIM1 protein regulates calcium levels in the ER and initiates the opening of calcium channels in the cell membrane, such as ORAI1, allowing calcium influx. Recently, SOCE has been shown to be crucial for the migration of cancer cells, including breast cancer cells [68]. Disabling key SOCE components, such as ORAI1 and STIM1, may reduce the ability of cells to migrate and metastasize. However, knocking down STIM1 in MDA-MB-231 breast cancer cells led to accelerated migration and increased cell invasiveness. Interestingly, the effect of STIM1 on migration was not solely the result of its role in SOCE. Further analysis showed that STIM1-knockdown cells had reduced NFAT1 levels, while the overexpression of STIM1 resulted in slower migration [68]. Therefore, STIM1 modulates NFAT1 expression and thus affects migration. On the contrary, ORAI1 knockdown resulted in slower cell migration compared to cells without mutation [68].

The inhibitor of apoptosis stimulating protein of P53 (iASPP)–transmembrane and coiled-coil domains 1 (TMCO1) axis plays a key role in modulating calcium homeostasis and the sensitivity of cancer cells to apoptosis. iASPP is often overexpressed in cancers, promoting drug resistance and a poor prognosis by stabilizing and activating antioxidant nuclear factor erythroid 2-related factor 2 (NRF2) independently of cellular tumor antigen p53 (P53) [69]. This reduces oxidative stress and contextually inhibits apoptosis [70]. Yet, the impact of iASPP activation on calcium signaling dysregulation in cancer cells remains unexplored. The TMCO1 calcium channel is degraded by the E3 ligase Gp78. However, the iASPP oncogene stabilizes TMCO1 by competing with Gp78 and preventing its degradation. Zheng et al. showed that inhibition of iASPP-TMCO1 expression increased cell sensitivity to calcium-induced apoptosis [14]. In contrast, TMCO1 overexpression resulted in the inhibition of apoptosis. The chelation of cytosolic calcium by BAPTA-AM, which binds to free calcium ions in the cytosol, abolished the effect of induced apoptosis, confirming the key role of cytosolic calcium in the apoptosis of these cells. The calpain inhibitor PD150606 also reduced the effect of iASPP silencing on apoptosis, suggesting that iASPP regulates calcium-dependent apoptosis by reducing calpain activity. These results highlight the importance of the iASPP-TMCO1 axis in the pathogenesis of colorectal cancer and the potential for therapies targeting calcium homeostasis in cancer treatment [14].

Li et al. investigated the mechanisms that lead to apoptosis, focusing on the role of calcium, the ER, and the C/EBP homologous protein (CHOP) pathway, along with its interaction with IP3Rs [54]. IP3Rs are critical intracellular calcium release channels, existing in three isoforms (IP3R1, IP3R2, and IP3R3) that regulate various cellular functions, including cell death. IP3Rs can interact with BCL-2 family proteins, key regulators of apoptosis, to modulate IP3-induced calcium release [71]. The interaction sites on an IP3R include the C-terminal region, the middle of the regulatory region, and the IP3-binding domain, each of which differentially affects calcium signaling and cell survival. These interactions have important implications for understanding cell death, cancer progression, and potential therapeutic targets. During ER stress, CHOP induces the expression of ER oxidase 1 alpha (ERO1-α) [54]. In turn, ERO1-α enhances IP3R activity, leading to increased calcium release from the ER into the cytoplasm [72]. This process is crucial for the induction of apoptosis under conditions of long-term or severe ER stress. Using siRNA and genetic knockout models, researchers demonstrated that the suppression of ERO1-α or IP3Rs attenuates apoptosis in macrophages subjected to ER stress inducers [54]. On the contrary, reconstitution of ERO1-α restores apoptosis in CHOP-deficient macrophages, demonstrating the dependence of this pathway on CHOP. Furthermore, the study indicates that the redox state of the ER regulates IP3R function through ERO1-α-mediated oxidation, which facilitates IP3R activation. This finding is supported by experiments using antioxidants that inhibit apoptosis in cells subjected to ER stress. In vivo studies showed that macrophages from insulin-resistant mice exhibit increased susceptibility to ER-stress-induced apoptosis [54]. These findings shed light on a novel pathway in which CHOP-induced ERO1-α increases IP3R activity, linking ER stress with calcium-ion-dependent apoptosis and providing new therapeutic targets for diseases involving ER-stress-induced cell death.

It is also worth examining the role of BCL-2 family proteins, which play a complex role in the regulation of calcium signaling and cell death, acting as both promoters of cell survival and inducers of apoptosis depending on the cellular context and protein interactions. One of the functions of BCL-2 family proteins is the regulation of the flow of calcium ions between cell organelles, including the ER and mitochondria [73,74]. By primarily controlling calcium homeostasis in the ER, BCL-2 prevents excessive calcium release into the cytoplasm and influx into mitochondria, which could lead to its overload and trigger apoptosis. The BCL-2 proteins bind to IP3Rs in the ER, modulating their activity and reducing calcium release into the cytoplasm [75]. Some members of the BCL-2 family, such as B-cell lymphoma-extra large (Bcl-xL), can bind directly to IP3Rs, stabilizing the receptor and reducing its activity, thus protecting the cell from calcium stress and death [55]. In contrast, pro-apoptotic proteins, including BAX and Bcl-2 homologous antagonist killer (BAK), promote calcium release from the ER. In the presence of different stressors, these proteins can oligomerize in the mitochondrial membrane, creating pores that lead to the release of cytochrome c and the activation of caspases, which are key to apoptosis [10]. Interactions between BCL-2 proteins and calcium signaling are particularly important in the context of cancer cells. Anti-apoptotic members of the BCL-2 family are often overexpressed in cancer cells, preventing apoptosis and promoting the survival of cancer-affected cells. The BCL-2 proteins also influence the functioning of other proteins responsible for the transport and release of calcium, such as SERCA. On the one hand, BCL-2 can support its activity by increasing its expression at mRNA and protein levels. Furthermore, it may bind to SERCA2, increasing its resistance to inhibition and supporting the maintenance of calcium homeostasis in the ER after extracellular calcium depletion [73]. However, in some cases, it may destabilize the protein, leading to inhibition of its activity [76]. By binding to IP3Rs, BCL-2 modulates their activity and reduces calcium release into the cytoplasm, preventing calcium stress. Membrane-localized voltage-dependent anion channels (VDACs), RyRs, and plasma membrane calcium ATPases (PMCAs) also interact with BCL-2, constituting a complex network of regulation of cellular calcium homeostasis [77,78,79,80]. Understanding these mechanisms may lead to the development of more effective anticancer therapies that target calcium signaling pathways regulated by BCL-2 proteins.

VDACs, located on the outer mitochondrial membrane, regulate calcium flux between the cytosol and mitochondria, influencing mitochondrial homeostasis and cellular fate [81]. Several compounds have been described that promote a high-conductance state of VDACs, facilitating increased calcium uptake, which can lead to mitochondrial calcium overload and the opening of the mitochondrial permeability transition pore (mPTP). Erastin, for instance, binds directly to VDAC-1, causing increased calcium influx and triggering mitochondrial dysfunction [82]. The opening of the mPTP disrupts mitochondrial membrane potential, resulting in the loss of ATP synthesis, organelle swelling, and the release of pro-apoptotic factors such as cytochrome c, ultimately leading to tumor cell death. Notably, normal epithelial cells with low levels of VDAC-1 are unaffected by erastin. In vivo studies demonstrated that erastin inhibited tumor growth in mice without significant toxicity, underscoring its therapeutic potential [81]. Keinan et al. observed that various apoptosis inducers—including staurosporine, curcumin, arsenic trioxide, etoposide, cisplatin, selenite, hydrogen peroxide, and UV light—induce VDAC-1 oligomerization via distinct mechanisms [83,84]. This oligomerization leads to the formation of channels in the mitochondrial membrane and the release of cytochrome c, which activates the caspase cascade, culminating in apoptosis. VDAC modulation, however, is not limited to pro-death outcomes. Compounds that stabilize the low-conductance state of VDACs can limit calcium influx, protecting mitochondria from overload and preventing cell death [81]. This dual functionality makes VDACs a desirable therapeutic target. Modulating VDAC conductance offers promising strategies to selectively induce apoptosis in cancer cells or protect healthy cells from excessive calcium-mediated stress. Further research into specific compounds and the molecular mechanisms regulating VDAC activity could enhance the understanding of its role in calcium signaling and mitochondrial function, paving the way for novel therapeutic approaches targeting mitochondrial dysfunction in cancer and other diseases.

VDAC disruption can have profound consequences for cellular homeostasis. Excessive calcium influx into mitochondria, facilitated by prolonged high-conductance states of VDACs, can trigger oxidative stress through increased ROS production, promoting mPTP opening and the release of apoptotic factors [81]. These events not only initiate apoptosis but also impair ATP synthesis, resulting in energy deficits and organelle dysfunction. Several compounds, such as sanglifehrin A, cyclosporin A, and bongkrekic acid, have demonstrated the potential to mitigate these effects by inhibiting mPTP opening [82]. Cyclosporin A, for example, binds to cyclophilin D, a key regulator of mPTP, stabilizing mitochondrial function and reducing calcium-induced damage [85]. 4-hydroxytamoxifen (4-OHT), the main active metabolite of tamoxifen, also exhibits the ability to modulate mitochondrial function, including protecting against mPTP opening. Studies have shown that 4-OHT protects mitochondria from lipid peroxidation induced by oxidative stress, stabilizes mitochondrial membrane potential, and limits calcium ion influx [86]. Although its protective action is weaker compared to cyclosporin A, 4-OHT demonstrates synergistic potential when combined with cyclosporin A, significantly enhancing mitochondrial protection [86]. Furthermore, 4-OHT can bind to the transmembrane protein adenine nucleotide translocator, suggesting a possible mechanism of its action as an mPTP inhibitor [87]. These protective properties may play an important role in preventing mitochondrial dysfunction in neurodegenerative diseases, where lipid peroxidation and mPTP induction are key contributors to cell death. The potential therapeutic applications of 4-OHT in oncology are also noteworthy. The anti-estrogenic effects of 4-OHT, combined with its ability to prevent mPTP opening, suggest that it could work synergistically with other compounds, such as acitretin, in the treatment of hormone-dependent cancers [86]. Studies of these combinations indicate the potential to enhance therapeutic efficacy by simultaneously targeting mitochondrial regulatory mechanisms and cancer cell proliferation pathways.

Pierro et al. presented the first evidence of the influence of oncogenic K-RAS on calcium signals in the context of neoplastic transformation [88]. By comparing isogenic colorectal cancer cell lines expressing one copy of the K-RASG13D/WT mutant, they found that deletion of K-RASG13D increased IP3-dependent calcium signals and calcium flux between the ER and mitochondria, which increased the sensitivity of cells to pro-apoptotic stimuli. K-RASG13D may inhibit IP3 signaling from the ER and mitochondrial calcium uptake, promoting cell survival in an oncogenic context [88]. Hedgepeth et al. analyzed the pro-apoptotic mechanism of action of BRCA1, identifying its physical and functional interaction with IP3R1 during apoptosis, enhancing IP3R activity [89]. The authors suggested that BRCA1 promotes cell death based on IP3R-dependent increases in calcium release. Another study explored chetomin’s potential anticancer effects on triple-negative breast cancer (TNBC) cells and the underlying molecular mechanisms [56]. TNBC, known for its poor prognosis and limited therapeutic options, is associated with several molecular targets, including DNA repair mechanisms such as the phosphoinositide 3-kinase (PI3K)/rapamycin-dependent protein kinase (mTOR) pathway, crucial for regulating cell growth and proliferation, and stress-related ER pathways like hypoxia-inducible factor 1 alpha (HIF1-α)/aryl hydrocarbon receptor nuclear translocator (ARNT), critical for the response to hypoxia. Chetomin has been found to inhibit activation of the PI3K/mTOR pathway, leading to ER stress and apoptosis in MDA-MB-231 and MDA-MB-468 cells. It also affects poly ADP-ribose polymerase (PARP) and HIF1-α/ARNT activity, potentially promoting DNA damage accumulation and apoptosis. Chetomin-induced mitochondrial dysfunction disrupts calcium homeostasis, causing cytochrome c release and caspase activation, thus triggering apoptosis through the intrinsic pathway [56].

In the context of calcium signaling and its role in initiating apoptosis in cancer cells, several key proteins and cellular pathways involved in these processes have been identified. These examples illustrate the diversity and complexity of mechanisms that regulate cell survival or death through interactions with calcium ions. We have summarized some of them in Figure 1. The importance of calcium in these processes is undeniable. However, ongoing research is essential to fully understand the mechanisms of calcium signaling and explore opportunities to target them in the development of cancer therapies.

The vulnerability of cancer cells to mitochondrial calcium overload is particularly significant in certain “hyperdependent” cancer models, such as acute lymphoblastic leukemia and acute myeloid leukemia. Cancer cells, particularly those in aggressive subtypes like T-cell acute lymphoblastic leukemia, rely heavily on mitochondrial function for metabolic processes like energy production, biosynthesis, and survival under stress [90]. Unlike healthy cells, which maintain stable calcium homeostasis, cancer cells often exhibit disrupted calcium signaling pathways, which they utilize for regulating their altered metabolic processes. This dysregulated calcium signaling in cancer cells plays a central role in their resistance to apoptosis, allowing them to survive even under conditions that would typically trigger cell death. Mitochondrial calcium dysregulation not only supports cell survival in these conditions but also contributes to the metabolic plasticity observed in cancer cells, such as the Warburg effect, where even in the presence of oxygen, cells favor glycolysis over oxidative phosphorylation [91]. For instance, in venetoclax-resistant leukemia stem cells (LSCs), increased mitochondrial calcium uptake supports their reliance on oxidative phosphorylation, enabling survival even in the presence of chemotherapy [92]. Targeting mitochondrial calcium uptake in these cells has shown promise as a therapeutic strategy, leading to the eradication of resistant LSCs. Similarly, in T-cell acute lymphoblastic leukemia, a highly aggressive subtype, mitochondria are critical for generating reactive oxygen species (ROS), calcium signaling, and the regulation of cell death pathways [90]. Mitochondria-targeting therapies, such as those that promote mitochondrial calcium overload or disrupt VDAC1, have been proposed as effective approaches [81,83,84]. By exploiting these mitochondrial vulnerabilities, new treatments could specifically target malignant cells while sparing normal tissues. Strategies that target mitochondrial calcium signaling and disrupt metabolic processes can selectively induce cell death in tumor cells while overcoming resistance mechanisms that often impair the effectiveness of conventional therapies.

While apoptosis is a hallmark of nucleated cells, its counterpart in enucleated cells, such as erythrocytes, is eryptosis. This form of RCD ensures the removal of damaged or dysfunctional red blood cells in a non-inflammatory manner. Eryptosis shares certain morphological features with apoptosis, such as cell shrinkage and membrane blebbing, but it relies on distinct mechanisms adapted to the unique structure of erythrocytes, which lack nuclei and mitochondria [93]. By exposing phosphatidylserine (PS) on the outer leaflet of the cell membrane, eryptotic erythrocytes signal for phagocytosis, preventing the release of inflammatory damage-associated molecular patterns (DAMPs). Calcium plays a pivotal role in regulating eryptosis, acting as the master initiator of the process. Elevated intracellular calcium levels activate calcium-sensitive Gardos channels, causing the efflux of potassium and chloride [94]. This ionic imbalance reduces intracellular osmotic pressure, causing water loss, and results in cell shrinkage, a hallmark of eryptosis. Simultaneously, calcium drives the activity of scramblase, which redistributes PS from the inner to the outer leaflet of the plasma membrane, marking the cell for clearance. Calcium also inhibits flippase, an enzyme responsible for maintaining PS on the inner membrane, further ensuring that PS remains exposed externally [93,95]. Additionally, calcium activates calpain, a cysteine protease that degrades cytoskeletal proteins, contributing to membrane blebbing and further facilitating cell disassembly [93]. The role of calcium in eryptosis is intricately linked to broader signaling networks. ROS often amplify calcium influx, creating a feedback loop that accelerates the process. Prostaglandin E2 and protein kinases such as protein kinase C (PKC), p38 mitogen-activated protein kinase (MAPK), and casein kinase 1α (CK1α) further modulate calcium entry, reinforcing its regulatory role [93,96]. The interdependence of these pathways underscores calcium’s importance not only as a direct effector but also as a coordinator of the eryptotic machinery. Intriguingly, despite its centrality, calcium is not indispensable in all cases of eryptosis. Alternative signaling molecules, including ceramide and ROS, can independently execute the process, demonstrating the adaptability of the eryptotic program to various cellular conditions [93,97,98]. The dominance of calcium in eryptosis has sparked interest in its potential relationship with calcicoptosis, a form of calcium-induced regulated cell death typically observed in nucleated cells. Unlike calcicoptosis, which often involves mitochondrial calcium overload, eryptosis relies on cytosolic calcium as its primary regulator. This distinction highlights the unique ways in which calcium drives regulated cell death in erythrocytes, raising questions about whether eryptosis represents a specialized form of calcicoptosis or an entirely distinct entity. Addressing this question will require further investigation into the molecular mechanisms shared by and unique to these calcium-dependent phenomena.

### 3.2. Calcium Overload in Necroptosis

Necroptosis is a regulated cellular process with a morphology similar to necrosis that also depends on calcium homeostasis. Unlike apoptosis, this process occurs independently of caspases [10]. It is induced in response to death stimuli such as tumor necrosis factor α (TNF-α) and Fas ligand (FasL). Upon binding of the ligand to the death receptor on the cell surface, receptor-interacting protein kinase (RIPK)-1 is deubiquitinated by cylindromatosis protein (CYLD) [99]. RIPK1 then forms a necrosome complex with RIPK3, and lineage kinase domain-like protein (MLKL). In cells lacking caspase 8, RIPK1 is activated by autophosphorylation of serine 161 [100]. The necrosome induces the production of ROS in mitochondria, which act as executioners, leading to cell death. During necroptosis, cells release DAMPs that promote an antitumor immune response [10]. In the case of death-receptor-dependent necroptosis, the increase in cytosolic calcium may result from the formation of the necrosome complex, which leads to the trimerization of MLKL and its translocation to the cell membrane [101]. There, MLKL interacts with the transient receptor potential melastatin 7 channel, enabling the influx of calcium into the cell and, through a feedback mechanism, intensifying necroptosis. Blocking the transient receptor potential melastatin 7 channel reduces calcium concentrations and prevents necroptosis [101]. Another study found that calcium signaling is a side effect of necrosome formation and may not be essential for cell death. However, in necroptosis that is independent of death receptors, such as that induced by viruses or drugs, calcium acts as a modulator of necrosome complex proteins.

Nomura et al. investigated the mechanism of necroptosis in human neuroblastoma cells infected with the hemagglutinating virus of Japan envelope that do not express caspase 8 [102]. They identified that the fusion of the virus with the cell membrane promotes an increase in the concentration of calcium ions in the cytoplasm. Calcium accumulation activates CaMKII, which then phosphorylates RIPK1. As a result, the RIPK1-RIPK3 complex led to necrosome formation and necroptotic death. This raises the possibility that the manipulation of calcium signaling through the use of calcium ionophores may represent a potential therapeutic strategy for the treatment of cancers resistant to traditional apoptotic therapies.

Mitofusin-2 (MFN2), belonging to the family of mitochondrial fusion proteins, regulates the transport of calcium ions from the ER to the mitochondria. MFN2 promotes the induction of cell death processes and DNA hypermethylation [59]. Its deficiency may cause mitochondria to become overloaded with calcium ions. Studies have shown that the kinase inhibitor sorafenib can induce cardiomyocyte necroptosis by disturbing calcium ion homeostasis [60]. Sorafenib alters the components of the mitochondria-associated ER membrane, which may be related to its ability to inhibit signaling pathways that regulate gene and protein expression, including MFN2. Reducing MFN2 activity disrupts the proper flow of calcium ions from the ER to the mitochondria, causing their overload. Excess calcium ions lead to CaMKII activation, which then leads to cell necroptosis. These assumptions were confirmed using KN93, a CaMKII inhibitor that reversed sorafenib-induced activation of the RIPK3/MLKL pathway [60]. This study showed that MFN2 is another key player in the balance between cell protection and cell death.

Mitochondria are organelles that play an important role in regulating both necroptosis and apoptosis. A commonly accepted hypothesis is that excess calcium in the mitochondrial matrix is one of the main factors activating mPTP, which disrupts their electrochemical gradient, leading to mitochondrial swelling and cell death. Several proteins are responsible for transporting calcium into mitochondria, including VDACs and mitochondrial calcium uniporter (MCU), which are regulated by mitochondrial calcium uptake proteins 1/2 (MICU1/2) and the essential MCU regulator (EMRE) [103]. MICU1/2 can bind to MCU and EMRE, forming a complex that limits the influx of calcium ions into the mitochondria. Under conditions of increased calcium concentration in the cytoplasm, MICU1/2 change their conformation, leading to the opening of the MCU channel and the influx of calcium into the mitochondria. Additionally, EMRE integrates the function of the MCU with MICU1/2, influencing the activity of the entire complex. An increase in mitochondrial calcium concentration affects the conformation of proteins that form the PTP, leading to its opening. The mPTP is composed of several proteins, including VDACs, cyclophilin D, adenine nucleotide translocator, and others [104]. The opening of the PTP results in a sharp increase in the permeability of the inner mitochondrial membrane. Consequently, there is a loss of membrane potential due to depolarization of the mitochondrial membrane, which disrupts ATP production [104]. This is accompanied by the release of components from the mitochondrial matrix, including pro-apoptotic factors such as cytochrome c. Open PTPs also increase ROS production, creating a positive feedback loop that exacerbates mitochondrial damage. As a result, mitochondria are irreversibly damaged, and necroptotic mechanisms are triggered.

Studies have shown that decreased MCU expression can result in decreased calcium influx into mitochondria [105,106]. The calcium concentration then remains at a level insufficient to activate mPTP, as a result of which the mitochondria are not subject to damage induced by calcium stress. The lack of mPTP opening does not cause a loss of membrane potential, impaired ATP production, or release of pro-apoptotic factors. However, MCU knockout does not necessarily prevent cell death, because cells can still undergo necroptosis through mechanisms related to accumulation of cytoplasmic calcium and/or induction of alternative cell death pathways. In the studies of Kwong et al. and Luongo et al. on MCU-knockout animals, no increased survival was observed compared to wild-type animals during ischemia–reperfusion injury [107,108]. Although mitochondria showed impaired calcium uptake, it was not completely abolished. Other studies have shown that despite altered calcium signaling and impaired pyruvate dehydrogenase function, mPTP opening was reduced, which correlated with reduced mitochondrial calcium uptake, but cell death was comparable to controls [109]. Interestingly, non-knockout cells responded to cyclosporine A (mPTP inhibitor) and ruthenium red (MCU uniporter inhibitor), which reduced mPTP opening, whereas knockout cells did not [109]. Blocking calcium ion influx into mitochondria can lead to a rapid increase in cytosolic calcium concentration, which activates mitochondrial shape transition (MiST) and/or alternative cell death pathways, including non-mitochondrial necroptosis [110]. MiST is a recently identified phenomenon that occurs in the cytoplasm of cells in response to calcium stress. It involves changes in mitochondrial morphology that are not associated with mPTP opening or swelling. MiST is triggered by persistently high levels of calcium ions in the cytoplasm, rather than by their uptake through uniporters such as MCU. One of the key proteins involved in this process is the mitochondrial Rho GTPase 1 (MIRO1) protein associated with the mitochondrial outer membrane, which acts as a calcium sensor. By binding to calcium ions, MIRO1 disrupts the MIRO1/kinesin family member 5B/tubulin complex, leading to impaired mitochondrial trafficking and morphological changes [110]. MiST is also thought to be involved in necroptosis and other forms of cell death by disrupting normal mitochondrial functions, including ATP production, oxidative stress management, and regulation of membrane potential.

As previously mentioned, MICU1 is an important player in calcium-ion-induced necroptosis, which not only regulates MCU but also plays a role in cell death. Reduced levels of the MICU1 protein in cancer cells have been shown to lead to increased mitochondrial calcium levels and increased ROS production, which in turn contributes to necroptotic death [111]. Similarly, increased calcium levels caused by SOCE activation in hepatocytes and cancer cells are associated with elevated ROS production and a higher rate of cell death in MICU1-knockout cells [112]. Although blocking the transportation of calcium through the calcium uniporter to mitochondria has an ambiguous effect on necroptosis, MICU1-knockout cells are distinctly more susceptible to cell death due to mitochondrial calcium overload. Antony et al. also reported that MICU1-KO mice were perinatally lethal, whereas MCU-KO mice survived [113]. In addition to the proteins discussed above, there are other players involved in the detection or regulation of calcium ion levels, their transport between the ER, cytoplasm, and mitochondria, and the induction of necroptotic death. Their detailed discussion is beyond the scope of this review, but we recommend other articles [104,114,115].

In summary, necroptosis can occur both in a death-receptor-dependent and -independent manner (Figure 2). In the first case, the increase in calcium concentration in the cytoplasm is mainly the result of the activation of the necrosome complex. In turn, in death-receptor-independent necroptosis, calcium acts as an exit factor that modulates proteins of the necrosome complex and thus leads to cell death.

### 3.3. Calcium Overload Leads to Necrosis

Necrosis is a form of accidental cell death (ACD) that occurs due to mechanical damage or severe pathological stress, lacking the regulatory machinery characteristic of RCD. It is characterized by the rapid disintegration of cellular structures, leading to the release of their contents and the development of an inflammatory reaction [116]. Necrosis can also be induced by calcium ion overload [57,58]. Calcium regulates many cellular processes that can be impaired in cancer cells. In some types of cancer, an increase in the expression of specific calcium-permeable ion channels is observed [91,117]. Silencing or pharmacological inhibition of the overexpression of these channels can affect the proliferation and metastasis of cancer cells [57,58]. Alternatively, overexpression of ion channels can lead to cancer cell death.

In a study, Wu et al. used the MCF-7 cell line with inducible overexpression of transient receptor potential cation channel subfamily V member 1 (TRPV1) to assess the role of TRPV1 levels in cell death mediated by the TRPV1 activator [118]. TRPV1 activation in MCF-7 cells led to increased intracellular calcium levels, which induced cell death, mainly through a necrotic mechanism. This was confirmed by the increase in necrotic markers after TRPV1 activation, such as RIP3, which confirmed the nature of cell death. This effect depended on both the level of channel expression and the activator concentration. Higher TRPV1 expression increased the sensitivity of cells to the activator and triggered cell death. Moreover, the addition of a calcium chelator (BAPTA) protected cells from necrosis, confirming that the observed effect was dependent on the presence of calcium ions. Importantly, TRPV1 activation did not promote MCF-7 cell proliferation, thus not posing a risk of tumor progression [118]. Therefore, activation of overexpressed calcium-permeable ion channels may be an alternative therapeutic strategy in the treatment of tumors with defects in apoptotic pathways.

Targeting calcium channels may be an effective therapeutic strategy. Recently, calcium channels such as the transient receptor potential calcium channel C4 (TRPC4) and C5 (TRPC5) have been reported to be modulated by the small-molecule sesquiterpene natural activator (−)-englerin A [57]. In studies on renal cell carcinoma cells, (−)-englerin A increased calcium ion influx into the cells, leading to their overload and death. The cytotoxic effect of the compound occurred within a few minutes, which excluded the regulation of gene transcription as a mechanism. There were also no changes in PKC-theta expression, which was previously assumed to be the primary mechanism of action of the compound. Finally, the sesquiterpene derivative was shown to directly affect calcium channels, promoting ion influx. (−)-englerin A showed a higher efficacy in inhibiting the growth of renal cell carcinoma cells compared to other types of cancer [89]. In another study, (−)-englerin A induced the cell death of renal cell carcinoma cells by necrosis, without posing damage to normal cells [58]. (−)-englerin A increased intracellular calcium levels, leading to increased ROS production and activation of necrotic signaling. A four-fold increase in calcium ion concentration was observed after treatment with (−)-englerin A compared to the effect of ionomycin (calcium ionophore). However, no increase in the expression of markers for apoptosis, autophagy, or pyroptosis was found, clearly indicating a necrotic mechanism of cell death [58].

Electroporation is one of the techniques used in cell biology, cancer, and gene therapy which allows the introduction of substances into cells through a temporal increase in the permeability of cell membranes [119]. This is achieved through the application of an electric field that creates channels in the cell membrane, allowing molecules such as drugs or nucleic acids to pass through (Figure 3). Studies on the anticancer effects of calcium have shown that calcium electroporation can be an effective therapeutic method. Calcium electroporation led to a decrease in cancer cell survival in vitro (EC50 = 0.57 mmol/L calcium) [120]. Moreover, studies in animal models observed the elimination of almost 90% of ulcerative tumors, which was accompanied by a decrease in ATP levels [120]. Histological analyses revealed the presence of necrotic areas. The effectiveness of calcium electroporation was attributed to the depletion of cellular energy resources and an increase in intracellular calcium levels, which consequently led to necrotic cell death [120]. The advantages of calcium electroporation therapy include high precision in molecule delivery and increased cell sensitivity, translating into higher efficacy in tumor eradication, as well as a reduced risk of side effects and systemic toxicity.

Ji et al. studied the effect of stemphol (STP), a fungal compound derived from resorcinol, on cancer cells [121]. STP has been shown to cause cancer cell death via noncanonical, caspase-independent pathways, including necrosis, which is triggered by the dysregulation of ROS levels and calcium homeostasis. The primary mechanism involves the release of calcium ions from the ER into the cytoplasm and their accumulation in mitochondria, leading to the overload of these organelles and the opening of the mPTP, resulting in swelling and mitochondrial rupture. The rupture of mitochondria and the release of their contents induce necrosis. Studies in leukemic models have demonstrated that STP can induce cell death by causing mitochondrial stress and releasing markers associated with immunogenic cell death, which activate the immune response against cancer cells. The high mobility group box 1 (HMGB1) protein was released from cells undergoing necrosis, acting as one of the DAMPs that stimulate the immune response. Its release was associated with the activation of macrophages and dendritic cells in the tumor microenvironment. Importantly, STP induced the release of HMGB1 protein from leukemic cells independently, which may enhance its effectiveness as an inducer of the immune response against tumor cells. In silico analyses also indicated that STP has favorable pharmacological properties, making it a potential candidate for anticancer therapy, especially against tumors resistant to apoptotic death [121].

Metabolic dysfunction is a key aspect of cancer cell initiation, progression, and resistance. Abnormal lipid accumulation, altered metabolism, and autophagy contribute to a high-energy state and therapeutic failure. Rupert et al. identified the lysosomal purinergic receptor 4 (P2XR4) as a crucial regulator of endothelial cell motility and mitochondrial dynamics [122]. Its inhibition has been shown to affect cell survival in clear-cell renal cancer by disrupting calcium metabolism and homeostasis. P2XR4 influences lysosome–mitochondria interactions, regulating oxidative phosphorylation and energy flux. Targeting P2XR4 may represent a promising therapeutic approach for tumors with high mitochondrial activity.

### 3.4. Calcium Overload in Other Types of Cell Deaths

Although apoptosis, necrosis, and necroptosis are among the most well-known and widely studied forms of cell death, calcium overload also plays an important role in less common forms, such as ferroptosis, pyroptosis, paraptosis, and others. These mechanisms, though less familiar, are crucial in the pathogenesis of neurodegenerative, neoplastic, and other diseases and are the focus of intensive scientific research.

Ferroptosis is a form of cell death dependent on iron ions, caused by the accumulation of lipid peroxides in cell membranes, leading to the loss of their integrity. Glutathione peroxidase 4 (GPX4) plays a key role in its regulation by neutralizing these peroxides. Its deficiency or dysfunction results in uncontrolled oxidative stress and cell death. Due to its specific mechanism involving iron metabolism and lipid oxidation, ferroptosis most commonly occurs in degenerative, neurodegenerative, or neoplastic diseases, where these factors are central to pathogenesis. Osteoarthritis, one of the most common degenerative diseases, is characterized by the gradual destruction of joint cartilage. The piezo type mechanosensitive ion channel component 1 (Piezo1) ion channel, which is activated in chondrocytes in response to mechanical stimuli, has been shown to increase calcium ion influx, leading to mitochondrial damage and elevated ROS production due to calcium imbalance [123]. This results in reduced glutathione production and decreased GPX4 levels, which regulate ferroptosis. Inhibition of Piezo1 with an inhibitor (GsMTx4) reduced calcium ion influx, ameliorating ferroptosis symptoms and improving chondrocyte metabolism [123]. In studies on lung cancer cells, the natural plant compound erianin induced ferroptosis, which was associated with the induction of oxidative stress, lipid peroxidation, and reduced glutathione levels [124]. The compound activated a calcium-calmodulin-dependent signaling pathway, leading to increased ROS production and iron levels, driving cell death. Blocking this pathway significantly reduced erianin-induced ferroptosis, confirming the key role of calcium ions in the ferroptosis process [124].

Recently, another study highlighted the importance of calcium in the mechanism of palmitic acid-induced ferroptosis in colon cancer cells [125]. The compound induced ER stress, leading to the release of calcium ions into the cytoplasm, where their increased concentration promoted endocytosis and transferrin recycling, resulting in iron ion accumulation. Iron overload and ROS-induced lipid peroxidation led to ferroptosis [125]. Therefore, certain compounds can induce cell death by activating ER stress associated with calcium ion release and TF-dependent ferroptosis.

Another newly identified type of programmed cell death is pyroptosis, which is associated with the development of an inflammatory response. It occurs when the activation of inflammasomes, such as NACHT, LRR, and PYD domains-containing protein 3 (NLRP3), leads to the activation of caspase 1 [10,126]. Caspase 1 then cleaves pro-inflammatory cytokines and gasdermin D. Cleaved fragments of gasdermin D create pores in the cell membrane, causing damage and inducing cell lysis, which leads to the release of cytoplasmic contents and promotes inflammation. We have recently discussed the mechanism of pyroptosis in our previous review. Therefore, this topic will not be described here in detail [10]. Calcium mobilization appears to be an important event in activating the NLRP3 inflammasome. This mobilization can occur as a result of stimuli that initiate the opening of channels in the cell membrane, allowing the influx of calcium ions from the external environment or promoting the release of calcium from the ER [127]. The efflux of potassium ions from the cell can be coordinated with the release of calcium from the ER and the opening of calcium channels in the plasma membrane. Factors such as nigericin, alum, sodium urate crystals, and the membrane attack complex have been shown to depend on calcium flux and potassium efflux to activate the NLRP3 inflammasome [127]. However, the exact role of calcium in NLRP3 activation is not yet fully understood. The complexity and interconnection of signaling pathways involved in inflammasome activation is a subject of ongoing debate in the scientific community. Recent reports indicate that cucurbitacin B exerts antitumor activity by inducing pyroptosis in non-small-cell lung cancer cells [128]. Cucurbitacin B acts by directly binding to toll-like receptor 4 (TLR4), which activates the NLRP3 inflammasome, leading to gasdermin D activation and pyroptosis. The compound also increases ROS production and releases calcium ions, which promote the cell death process. Deactivation of TLR4 reduced the anticancer effects of cucurbitacin B, such as increased ROS and calcium ion levels. The role of calcium in the context of pyroptosis was also examined by Pang et al., who studied vascular calcification in patients with chronic kidney disease [61]. Calcification is the result of metabolic disorders in vascular smooth muscle cells, which are associated with calcification and further intensify pyroptosis processes, contributing to the development of cardiovascular diseases. Studies tested the effect of irisin, a myokine secreted by skeletal muscles, which can protect against calcification associated with chronic kidney disease. Irisin inhibited calcium deposition, leading to reduced vascular calcification, decreased ROS levels, and lower NLRP3 activity [61]. Additionally, it induced autophagic mechanisms, supporting the removal of damaged organelles and proteins, and reduced the risk of oxidative stress. Controlling calcium levels and linking them to pyroptosis is an important direction in the therapy of diseases associated with inflammation and vascular calcification.

Autophagy is a process in which cellular components, such as damaged organelles, improperly formed proteins, or cellular debris, are degraded and recycled, allowing for the maintenance of homeostasis and survival under stress conditions. Calcium ions constitute one of the regulators and mediators of this process. Their intracellular concentration is strictly controlled and influences the activation of signaling pathways that direct autophagy. An increase in calcium ion concentration can activate extracellular signal-regulated kinase (ERK), which promotes autophagy [129]. In response to cellular stress, autophagy can protect cells from programmed death by removing damaged or dangerous cellular elements. Calcium serves as a regulator between apoptosis and autophagy, with cellular effects depending on the context rather than solely on the concentration of calcium ions. Calcium flux across the ER and mitochondria, regulated by calcium pumps and release channels such as PMCAs, SERCA, IP3Rs, and RyRs, determines whether cells survive through autophagy or succumb to apoptosis [62]. Studies by Zhou et al. demonstrate that cadmium disrupts calcium homeostasis by inhibiting calcium pump activity and activating calcium release channels, resulting in a dramatic intracellular calcium increase [62]. Cadmium affects the activation of RyRs and IP3Rs, stimulating the release of calcium stored in cellular reserves. CaMKII is activated by an increase in calcium concentration, which binds to calmodulin, and upon binding, it is transformed into its active form, allowing it to phosphorylate target proteins. The activation of CaMKII can stimulate the MAPK and mTOR signaling pathways, which are critical for balancing autophagy and apoptosis [130,131]. Recent research underscores the lysosome’s role as a key regulator of calcium-dependent autophagy pathways [132]. Calcium released through the lysosomal calcium channel mucolipin 1 activates calcineurin, a phosphatase that dephosphorylates transcription factor EB (TFEB), a major transcription factor for autophagy and lysosomal biogenesis. Dephosphorylated TFEB moves to the nucleus, enhancing the expression of autophagy-related genes [133]. Blocking calcineurin, either genetically or pharmacologically, inhibits transcription factor EB activation under stress, while its overexpression boosts autophagic responses [132]. This mechanism highlights how lysosomal calcium signaling connects cellular stress responses to autophagy regulation. The dual role of calcium in autophagy and apoptosis highlights its importance in determining cell fate. Calcium-dependent activation of calcineurin and its ability to regulate TFEB not only link calcium signaling to autophagy induction but also establish a broader role for calcium in maintaining cellular homeostasis [132]. Under conditions of prolonged stress, calcium dysregulation can shift autophagy from a protective mechanism to one that contributes to cellular dysfunction, depending on the severity and duration of the calcium imbalance.

Paraptosis is a type of cell death that is significantly different from apoptosis or necrosis. The process is characterized by the massive enlargement of cytoplasmic vacuoles and alterations in the structure of organelles [134]. An important element of paraptosis is the dysregulation of calcium management within the cell, particularly involving mitochondria and ER. Cannabinoids, including cannabidiol (CBD), can affect calcium balance in the cell by increasing its flow from the ER to mitochondria [135]. Excess calcium ions in the mitochondria lead to mitochondrial overload and swelling, which is a crucial stage of paraptosis. Disruption of proper mitochondrial function lowers the membrane potential and destabilizes the structure of the cristae, resulting in further cell damage. Concurrently, an increase in mitochondrial calcium levels depletes calcium resources in the ER, causing stress. This is evidenced by the increased expression of glucose-regulated protein 78 and CHOP proteins, which are involved in managing excess misfolded proteins [135]. Cannabinoids have been shown to modulate VDACs by increasing calcium permeability. This modulation exacerbates mitochondrial calcium overload and contributes to increased paraptosis, as observed in MCF7 breast cancer cells [135]. Other studies have demonstrated that the natural plant compound α-hederin induces paraptosis in colon cancer cells, which is associated with the activation of G-protein-coupled receptors and calcium signaling pathways [136]. α-Hederin stimulates calcium release from the ER through the activation of IP3Rs, leading to the activation of the PKC-alpha and MAPK cascade. Increased intracellular calcium contributes to the formation of vacuoles and enlargement of the ER and mitochondria, leading to paraptosis. This was confirmed by the inhibition of the Na^+^/Ca^2+^ exchanger, which effectively prevented the formation of vacuoles in cells treated with α-hederin, emphasizing the indispensability of calcium ions [136]. Xue et al. evaluated the effect of flavonoid morusin on epithelial ovarian cancer both in vitro and in vivo [137]. This compound effectively inhibited the proliferation and survival of cancer cells while reducing tumor growth. Morusin induced paraptosis-like cell death, which was manifested by increased calcium levels in mitochondria, accumulation of ER stress markers, ROS production, and loss of mitochondrial membrane potential. The application of a VDAC inhibitor blocked the influx of calcium into the mitochondria and prevented morusin-induced cell death [137]. The involvement of calcium that leads to overload and dysfunction of cellular structures appears to be an essential element of paraptosis.

As evidenced by the examples above, calcium ions play a key role as regulators of cell death mechanisms, including apoptosis, necrosis, necroptosis, and other less common types of RCD. Their participation in signaling between organelles determines the fate of the cell, influencing its survival or programmed death. The regulation of calcium signaling in cells is a valuable tool in the therapy of many diseases, including cancer, inflammatory disorders, and neurodegenerative diseases. The precise control of calcium levels can (a) activate cell death pathways in cancer cells, limiting tumor development and increasing the effectiveness of treatment while minimizing damage to healthy tissues; (b) reduce inflammation and protect cells from degeneration, thereby slowing disease progression; and (c) prevent excessive ion influx into cells, protecting them from damage or toxicity. It should also be noted that calcium ions play an important role in other types of cell death that are not covered in this review, highlighting their widespread occurrence, versatility, and importance in cellular processes [138,139,140,141].

### 3.5. A Switch Between Cell Survival and Death Pathways

Calcium ions play a pivotal role in determining cell fate, acting as a versatile switch between different forms of RCD, including apoptosis, necrosis, autophagy, and emerging forms such as calcicoptosis and pyroptosis. This dynamic interplay is context-dependent and influenced by the amplitude, localization, and temporal characteristics of calcium signaling within cellular compartments. In apoptosis, mitochondrial calcium overload often serves as a key initiator. Elevated mitochondrial calcium triggers the opening of the mPTP, leading to the release of pro-apoptotic factors such as cytochrome c, activation of caspases, and eventual cell dismantling [81,82]. Conversely, in necroptosis, cytosolic calcium may accumulate via TRP channels, enhancing the activity of necrosome components like RIPK1 and RIPK3 and amplifying cell death signaling via ROS generation and membrane disruption [60,115,118]. While apoptosis is characterized by caspase activity, necroptosis progresses independently of these proteases, demonstrating how calcium’s role can diverge to influence distinct death modalities [10,56,81,106,108]. Calcium signaling is also intricately tied to autophagy, acting as a promoter and regulator. For instance, calcium release from lysosomal stores through mucolipin 1 can activate calcineurin, leading to TFEB dephosphorylation and autophagy-related gene expression induction [132,133]. However, under persistent stress, calcium-mediated pathways may pivot autophagy from a protective mechanism to one that exacerbates cellular dysfunction, pushing the cell toward apoptosis or necrosis [129,131,132]. Emerging forms of RCD, such as pyroptosis and ferroptosis, further underscore calcium’s versatile regulatory role. In pyroptosis, calcium influx through channels like ORAI1 modulates inflammasome activation, facilitating gasdermin-dependent membrane pore formation [61,127]. Similarly, in ferroptosis, calcium promotes ER stress and lipid peroxidation, accentuating oxidative damage [125]. Despite its central role in these processes, calcium signaling and the mechanisms governing its influence on RCD are not yet fully understood. The identification of shared molecular players and pathways that determine the switch between different RCD types remains a significant challenge. Current evidence suggests that calcium drives the most likely form of cell death available within a given cellular context, prioritizing pathways based on the availability of molecular machinery and environmental conditions. This crosstalk highlights calcium’s role as a master regulator of RCD, with its impact determined by cellular context and the interplay between signaling networks. Targeting calcium homeostasis offers a promising avenue for cancer therapy, as selective modulation of calcium-dependent pathways could exploit the vulnerabilities of cancer cells while sparing normal tissues. Future research should focus on unraveling the precise molecular mechanisms by which calcium switches between death pathways, potentially paving the way for more effective therapeutic interventions.

## 4. Novel Calcium-Based Nanoparticles in Cancer Therapy

As the knowledge of calcium signaling deepens, there has been a growing trend toward the design of calcium-based nanoparticles that target cancer. The modulation of calcium levels in cancer cells may be an effective tool in inducing RCD and disrupting other key cellular processes, making calcium an attractive therapeutic element. For this reason, scientists, drawing on the achievements of nanotechnology, have started to develop innovative calcium-based drug carriers that can precisely deliver therapeutic doses of calcium directly to cancer cells. Below, we will discuss examples of such nanoparticles, their mechanisms of action, and the potential benefits and challenges of their use in cancer therapy.

Undoubtedly, one of the pioneers in the use of calcium-based nanoparticles in anticancer therapy was the aforementioned team of Zhang et al. [7]. They not only defined a new form of cell death but also demonstrated that calcium peroxide-based nanoparticles coated with sodium hyaluronate (SH-CaO_2_) can be used effectively in targeted anticancer therapy. In response to the acidic tumor environment, these innovative nanoparticles released calcium ions and hydrogen peroxide, leading to calcium overload and oxidative stress in tumor cells. This resulted in a significant reduction in tumor size and induced tumor cell death, confirming their therapeutic efficacy.

Li et al. developed an innovative molecule using amifostine to bind herceptin to amphiphilic nanoparticles of gelatin (AG)–iron oxide and calcium phosphate (CaP) [142]. This AGIO@CaP-CD system contained a pH-sensitive CaP shell and a degradable AG core, enabling the sequential release of hydrophobic curcumin and hydrophilic doxorubicin. Their dual targeting strategy with HER-AGIO@CaP-CD, integrating bioligand and magnetic targeting, significantly enhanced uptake in receptor tyrosine-protein kinase erbB (HER2)-overexpressing SKBr3 cells. The combination of herceptin, iron oxide nanoparticles, hydrophobic curcumin, and doxorubicin in this platform not only provided dual targeting capabilities but also synergistically induced apoptosis. The study highlighted the critical role of calcium in optimizing the therapeutic efficacy of this bidirectional co-delivery system, suggesting its potential as a robust tool for the treatment of HER2-positive cancers in both experimental and clinical contexts.

Another example is the development of calcium carbonate (CaCO_3_)-based nanoparticles designed as drug delivery systems to support anticancer therapy. These nanoparticles were used to carry kaempferol-3-O-rutinoside (KAE), a natural compound with potent anticancer properties that also disrupt calcium homeostasis in cancer cells. The CaCO_3_@KAE nanoparticles were synthesized by a gas diffusion reaction, in which calcium chloride anhydrate (CaCl_2_) and KAE were dissolved in ethanol and then converted into nanoparticles under vacuum conditions [143]. Additionally, the nanoparticles were modified by coating them with a cancer cell membrane (M@CaCO_3_@KAE), enabling better tumor targeting due to their immune evasion properties and homologous aggregation ability. Once delivered to the tumor, M@CaCO_3_@KAE degraded under the influence of the acid microenvironment, releasing KAE and calcium ions. KAE disrupts calcium balance in cancer cells, leading to calcium overload and programmed cell death. Calcium released from CaCO_3_ improved the effect of KAE-induced calcium overload, destroying mitochondrial structures, oxidative stress, and apoptosis [143]. The developed M@CaCO_3_@KAE nanoparticles therefore demonstrated therapeutic potential in cancer treatment due to the synergistic effects of KAE and calcium ions. Their use overcomes the limitations associated with the low solubility and bioavailability of KAE while ensuring effective and safe drug delivery directly to cancer cells.

In 2022, Li et al. developed nanoparticles for targeted anticancer therapy, with the primary goal of improving pyroptosis and thus increasing the elimination of cancer cells [144]. Researchers created nanoparticles composed of a nanopolymer that serves as a carrier for the ESCRT inhibitor and BAPTA-AM, a calcium ion chelator. Calcium is necessary for the activation of the ESCRT III complex, which is responsible for repairing cell membrane damage. The calcium chelator BAPTA-AM, contained within the nanoparticle, binds calcium ions, preventing their action and consequently inhibiting membrane repair. The nanoparticle structure was designed to enable the effective delivery of the inhibitor to cancer cells. The nanopolymer was synthesized using a method that ensures stability and controlled release of the inhibitor under appropriate intracellular conditions. When the cell membrane is damaged, calcium ions flow into the cell, activating the ESCRT III system responsible for membrane repair. BAPTA-AM, encapsulated in the nanoparticle, chelates calcium ions, preventing their influx into the cell and thus blocking the activation of ESCRT III [144]. This, in turn, prevents the repair of the cell membrane, enhancing pyroptosis and leading to cancer cell death. The nanoparticle structure can also include surface modifications that allow for the specific targeting of cancer cells. When used in combination with other therapies, the effectiveness of this approach can be increased.

One of the latest discoveries was the development of an advanced nanoparticle called a “Mito-Jammer”, a bimetallic nanoparticle designed to induce mitochondrial damage in cancer cells [141]. The nanoparticle enhances the cuproptosis process and induces an immune response, which leads to the inhibition of tumor growth and spread. The Mito-Jammer was synthesized in a “one-pot” process involving the incorporation of doxorubicin and calcium peroxide into hyaluronic acid-modified metal–organic frameworks, characterized by regular morphology and stability, as well as efficient loading of doxorubicin and calcium peroxide. Their surface was modified with hyaluronic acid to enable specific targeting of cancer cells dependent on aerobic respiration by recognizing the CD44 protein. In the tumor microenvironment, the nanoparticles disintegrate, releasing copper and calcium ions. The acidic environment causes calcium peroxide to generate hydrogen peroxide and calcium ions, which enhances the copper-dependent Fenton reaction and leads to the overproduction of ROS. At the same time, excessive ion accumulation leads to mitochondrial dysfunction, reduced ATP production, and increased oxidative stress [141]. All these processes enhance the cuproptosis effect, leading to cancer cell death. Additionally, the nanoparticle initiates immunogenic cell death, which supports the immune response against the tumor.

An equally innovative approach was the development of upconversion nanoparticles (UCNPs), capable of converting longer-wavelength light (near-infrared, NIR) to shorter-wavelength light [145]. These nanoparticles are coated with a zeolite framework of nitro-/nitrilo-imidazole (ZIF-82), which can release nitric oxide under the influence of NIR light. The method uses photothermal capabilities to control nitric oxide levels within cancer cells, which affects intracellular calcium stores and leads to their apoptosis. UCNPs were synthesized by co-precipitation and then coated with ZIF-82, which protects and controls the release of nitric oxide. The whole nanoparticle is stable and maintains its photothermal properties under the influence of infrared NIR light. When illuminated with NIR light, UCNPs convert energy, activating ZIF-82 to release nitric oxide, which opens the RyRs to the ER and causes the release of calcium ions into the cytoplasm. Excess calcium leads to tumor cell overload, mitochondrial dysfunction, and ultimately apoptosis [145]. These nanoparticles are intended for targeted cancer therapy, especially in cases where cancer cells overexpress ryanodine receptors. The therapy can be used in cancers that are resistant to other forms of treatment, as a support or alternative to standard therapies.

The development of nanoparticles directly based on calcium or indirectly using its mechanism of action as a key transmitter is an important step toward modern anticancer therapies. The role of calcium ions in RCD makes them an attractive therapeutic element. Designing nanoparticles that can precisely deliver calcium directly to cancer cells allows for modulation of its level in cells and can lead to its death through various mechanisms, such as calcium overload or oxidative stress. The discussed therapeutic strategy seems to be an interesting and effective approach because it allows a targeted and controlled release of calcium, increasing the chance of therapy success and minimizing side effects. These nanoparticles can also cooperate with other medicinal compounds, which gives a synergistic effect and allows drug resistance to be overcome. As a result, an innovative approach to cancer therapy based on the use of calcium can lead to more effective and specific methods for treating different types of cancer, offering new hope for effective therapies. Many nanoparticles have been developed to date that utilize calcium ion transmission mechanisms. Some of them are described in detail and summarized in Table 1. We also included other examples of nanoparticles that we did not describe, as this is not the main topic of our review.

In line with this evidence, several clinical trials were performed/are planned to utilize calcium electroporation as a therapeutic option. The NCT04958044 study aimed to establish the safety of the procedure in the treatment of esophageal cancer. Conducted at Copenhagen University Hospital Rigshospitalet in 2021 and 2022, the trial included patients with non-curable esophageal cancer. The procedure involved intratumoral injection of calcium gluconate, followed by the application of reversible electroporation using an endoscopic electrode, all performed in an outpatient setting. Eight patients were enrolled and treated in this study. The safety profile of calcium electroporation was generally favorable, with one serious adverse event reported (anemia, which required a single blood transfusion) and three minor adverse events (two cases of mild retrosternal pain and one case of oral thrush). Of the six patients who initially presented with dysphagia, two reported an improvement in their swallowing difficulties, while four experienced no change. Imaging evaluations revealed that one patient achieved a partial response, three patients showed no response, and four exhibited disease progression. Notably, six months post-treatment, the patient who responded well continued to maintain good health and did not require further oncological interventions. These findings suggest that calcium electroporation is a feasible and safe therapeutic option for esophageal cancer, with minimal side effects observed in this small cohort. The study lays the groundwork for larger trials to further evaluate the potential of calcium electroporation in achieving tumor regression and providing symptom relief in patients with esophageal cancer [149].

A phase I study (NCT03542214) was designed to explore the safety of endoscopic calcium electroporation as a treatment for colorectal cancer. Six patients with inoperable rectal and sigmoid colon cancer, all presenting with local symptoms such as bleeding, pain, and stenosis, were included in the study. The patients underwent endoscopic calcium electroporation and were subsequently monitored through endoscopy, computed tomography, and magnetic resonance imaging scans. Biopsies and blood samples were collected at baseline and at follow-up intervals of 4, 8, and 12 weeks post-treatment to assess histological changes and immune responses, specifically examining CD3/CD8 and programmed death-ligand 1 (PD-L1) levels through immunohistochemistry. Additionally, circulating cell-free DNA (cfDNA) levels were measured in blood samples to further evaluate the treatment’s impact. A total of 10 procedures were performed, and importantly, no serious adverse events were reported. Among the six patients, five reported relief from their local symptoms following treatment. Notably, one patient, who also received systemic chemotherapy, exhibited a clinical complete response. Despite these encouraging clinical outcomes, immunohistochemical analysis did not reveal significant changes in CD3/CD8 levels or cfDNA levels after treatment. This pioneering study demonstrates that calcium electroporation is a safe and feasible treatment option for colorectal cancer, with the potential to be administered as an outpatient procedure. This modality may offer significant benefits, particularly for fragile patients with limited treatment options, marking a promising step forward in the management of inoperable colorectal tumors [150].

The NCT01941901 phase I study was performed to evaluate the efficacy and safety of calcium electroporation compared to electrochemotherapy in patients with cutaneous metastases from breast cancer and malignant melanoma. The trial included seven patients with a total of 47 metastases. Out of these, 37 metastases were randomized and assessed for treatment response, while 10 were reserved for biopsy. This non-inferiority trial involved the individual randomization of metastases within each patient to receive either intratumoral calcium or bleomycin, followed by the application of electric pulses to the tumor site. Each metastasis was treated once, and the randomization code was revealed after a 6-month follow-up period. The objective response rate for calcium electroporation was 72%, with a complete response observed in 66% of metastases. In comparison, electrochemotherapy resulted in an objective response rate of 84% metastases, with a complete response in 68% metastases. There was no statistically significant difference between the two treatments, indicating that calcium electroporation is comparable in efficacy to electrochemotherapy. After one year, only 3 out of 25 treated metastases had relapsed, highlighting the durability of the response. Adverse events such as ulceration, itching, and exudation were slightly more frequent in metastases treated with bleomycin, and hyperpigmentation was exclusively observed in the bleomycin-treated group. These findings suggest that calcium electroporation is not only effective but also a potentially safer alternative to electrochemotherapy, with fewer skin-related side effects. This study supports the further investigation of calcium electroporation as a viable treatment option for patients with cutaneous metastases from various malignancies [151].

An exploratory phase I clinical trial (NCT03694080) was designed to assess the safety and efficacy of using calcium electroporation as a downstaging and immune-response-enhancing treatment in patients with early colorectal cancer before intended curative surgery. The study aimed to recruit a total of 24 patients, consisting of 12 patients with rectal cancer and 12 patients with sigmoid colon cancer, all of whom have histologically confirmed diagnoses and no indication for neoadjuvant chemoradiotherapy, whether experimental or standard-care-based, before surgery. The intervention will be closely monitored through a series of clinical examinations, blood sample analyses, and biopsies, along with patient-reported outcomes collected via questionnaires. These measures will help evaluate the safety of calcium electroporation, as well as its impact on tumor response and the associated immunological changes. By investigating these parameters, the study seeks to determine whether calcium electroporation can effectively downstage tumors and stimulate an enhanced immune response, potentially improving surgical outcomes and long-term prognosis for patients with early-stage colorectal cancer. Nevertheless, the status of the study is currently unknown (status for 25 August 2024) (study type—interventional; phase I; status unknown).

A similar phase I study is being conducted (NCT03051269) and is primarily aimed at investigating the safety of calcium electroporation in patients with recurrent head and neck cancers. This study also seeks to evaluate the tumor response using advanced imaging techniques such as PET/MRI (positron emission tomography/magnetic resonance imaging), along with clinical assessments and biopsies. Additionally, the trial includes a comparative analysis of calcium electroporation against electrochemotherapy to determine relative efficacy. To assess the impact of these treatments on patient quality of life, the study utilizes standardized questionnaires, including the EORTC QLQ C-30 and H&N35, which are specifically designed by the European Organisation for Research and Treatment of Cancer. These comprehensive evaluations aim to provide insights into both the clinical effectiveness and the patient-reported outcomes associated with calcium electroporation, potentially guiding its future application in treating recurrent head and neck cancers (study type—interventional; phase I, status unknown).

Recently, a phase II trial (NCT04259658) was completed, although the results have not yet been published. The study aimed to explore the histopathological mechanisms of tumor cell death in 24 patients with breast cancer metastases or other cutaneous or subcutaneous malignancies. In this non-randomized phase II study, the primary endpoint was to evaluate differences in tumor-infiltrating lymphocyte (TIL) populations in tissue samples from treated tumors, comparing samples taken two days after calcium electroporation treatment with those taken on the day of the procedure before treatment. The TIL content in biopsies was assessed through pathological examination and quantified as a percentage of the total cell population. Patients in this study were followed for up to three months, with biopsies collected at various time points depending on the number of tumors treated and whether one or two calcium electroporation sessions were administered. In addition to the primary focus on TIL populations, the study also aimed to analyze differences in tumor type, immune marker expression levels over time, vascular effects, and regressive changes. Moreover, the research included an examination of changes in systemic immunological markers, providing a comprehensive evaluation of the immune response following calcium electroporation. These findings are expected to shed light on the immunomodulatory effects of calcium electroporation and its potential to enhance antitumor immunity, which could have significant implications for the future of cancer therapy (phase II, interventional; completed).

A non-randomized phase II trial (NCT04225767) was designed to evaluate the clinical response rate of calcium electroporation treatment for malignant tumors of the skin and to assess its impact on the quality of life of patients. This study will be conducted in a real-world setting across three cancer centers in Northern Europe, focusing on patients with skin metastases and malignant wounds from various cancer histologies. A total of 30 patients will be enrolled, all of whom will have been offered standard care and other available treatment alternatives before participating in the trial. The protocol involves a single treatment session of calcium electroporation, with patients being followed through regular examinations for 12 months, beginning from the first day of treatment. The primary endpoint of the study is to evaluate the overall clinical response rate of calcium electroporation after two months. Additionally, a subset of patients will undergo MR scans post-treatment to further assess the therapeutic effects, while another subset will be interviewed to gauge the treatment’s impact on their quality of life. This study aims to provide crucial data on the effectiveness and tolerability of calcium electroporation in treating malignant skin tumors, potentially offering a new therapeutic option for patients with limited alternatives. The study is active but not enrolling participants (status for 25 August 2024) (study type—interventional; phase II; active, not recruiting).

Additional studies will evaluate the clinical safety and efficiency of calcium electroporation in early colorectal cancer (NCT03694080; phase I; unknown status), cutaneous and subcutaneous malignant tumors (NCT04225767; phase II; study active, not recruiting), and low-risk basal cell carcinoma (NCT05046262; phase III; unknown status) (Table 2).

Given the global burden of cancer, there is a pressing need for innovative and cost-effective treatments, particularly in low- and middle-income countries. Calcium electroporation emerges as a promising, cost-effective alternative that can be deployed in diverse healthcare settings, including hospitals and mobile outreach clinics, making it accessible across various income levels. The materials required—calcium, an electric generator, and electrodes—are affordable, widely available, and stable, facilitating their use in resource-limited settings [120]. Early clinical trials have demonstrated the safety and efficacy of calcium electroporation, showing its potential as a localized treatment that could be administered by a range of medical professionals, including surgeons, interventional radiologists, and even veterinarians. Moreover, its application does not necessitate the use of chemotherapeutic drugs, thereby reducing the cost and handling complexities associated with hazardous waste. Preclinical and clinical studies suggest that calcium electroporation might also trigger a systemic immune response, a promising area for further research. As an efficient, safe, and inexpensive treatment, calcium electroporation represents a valuable addition to cancer treatment strategies, offering hope for improved outcomes in both high- and low-resource environments [120].

## 5. Discussion

Calcium plays a crucial role in various physiological processes in the human body, impacting everything from muscle contraction and synaptic transmission to bone integrity and metabolic functions. Despite comprising about 1% of the body’s mass, calcium’s functions are vast and vital. As a universal messenger, calcium is crucial for regulating synaptic transmission, muscle contraction, heart rate, and blood clotting and ensuring the structural integrity of bones and teeth [20,21,29,33,34]. It is indispensable for hormone secretion and other metabolic processes [4]. Calcium is fundamental in the formation of hydroxyapatite, which, in conjunction with phosphate, creates a scaffold for connective tissue, providing mechanical strength and resistance to abrasion [29]. The precise regulation of calcium concentration is critical for cellular homeostasis, achieved through complex mechanisms involving pumps, channels, and exchangers, both in the cell membrane and the ER. Dysregulation of calcium levels can lead to severe conditions, including neurodegenerative diseases, pathological calcification, and rhabdomyolysis. For instance, calcium dysregulation in neurons is linked to the development of Alzheimer’s disease, while excess calcium deposition in tissues can lead to calcification and associated complications [27]. The complexity of calcium regulation highlights its crucial significance in health and disease, underscoring the need for further research to better understand and manage calcium-related disorders.

Calcium overload in cells is increasingly recognized as a critical factor in the development of metabolic and proliferative disorders [5,6]. Excessive calcium influx through ion channels or the release from intracellular stores, such as the ER and mitochondria, can lead to elevated intracellular calcium levels. This overload often results in oxidative stress, which further exacerbates cellular damage and dysfunction. Cancer cells exhibit a distinct vulnerability to mitochondrial calcium overload, arising from their dependence on mitochondrial metabolism for energy production and biosynthetic precursors [91]. Unlike healthy cells, which maintain strict calcium homeostasis, cancer cells often display altered calcium signaling and mitochondrial dynamics. This dependency is driven by metabolic adaptations, such as the Warburg effect—characterized by an increased reliance on glycolysis even in the presence of oxygen—and metabolic plasticity, which enables cancer cells to shift between oxidative phosphorylation and glycolysis based on environmental demands [152]. When calcium overload occurs in cancer cells, it triggers oxidative stress, increases ROS production, and promotes the opening of the mPTP, ultimately leading to cell death [64]. This heightened sensitivity contrasts with normal cells, which have more efficient calcium buffering mechanisms and pumps for regulating calcium levels and protecting against overload [153]. However, some researchers argue that cancer cells exhibit mechanisms that allow them to avoid calcium-induced apoptosis, making them less sensitive to calcium overload than healthy cells [154]. For instance, cancer cells may overexpress calcium-extruding proteins or suppress calcium entry into mitochondria, thus maintaining mitochondrial function and avoiding stress-induced apoptosis. These adaptations contribute to the survival and proliferation of cancer cells under conditions that would typically trigger cell death in normal cells. While cancer cells may employ strategies to mitigate calcium-induced stress, they remain more susceptible to certain forms of mitochondrial dysfunction due to their altered metabolic needs and reliance on mitochondrial function. This vulnerability provides an opportunity for therapeutic strategies that exploit calcium overload and mitochondrial dysfunction to selectively target cancer cells. Therapeutic approaches, such as VDAC modulation or mPTP activation, offer a promising way to selectively target cancer cells by pushing them beyond their metabolic thresholds, leading to energy crises and cell death, while sparing healthy cells that maintain more robust calcium regulatory systems [91].

Recent research has introduced the concept of ‘calcicoptosis’, a novel form of cell death characterized by calcium-induced damage and oxidative stress [7]. Innovations such as calcium peroxide nanoparticles have shown the potential to induce this type of cell death in cancer cells by leveraging their high calcium retention and oxidative stress. The synergy between calcium overload and oxidative stress presents a promising therapeutic strategy, particularly in cancer treatment, where targeting mitochondrial calcium dysregulation may enhance cell death and improve therapeutic outcomes.

Calcium overload plays a key role in many forms of RCD [54,55,56,57,58,59,60,61,62]. In apoptosis, excessive release of calcium from the ER into the cytoplasm triggers a cascade of events involving calpains, caspase activation, and mitochondrial changes [64]. This process is closely linked to ER stress and oxidative stress, which disrupt cellular homeostasis and lead to cell death. Additionally, calcium microdomains and signaling pathways, such as those involving calmodulin and NFAT1, regulate gene expression and cellular responses to stress [68]. In turn, calcium influx through channels such as TRPM7 and interactions with the necrosome complex can modulate necroptosis [101]. Necrosis, characterized by rapid cell disintegration and inflammation, can be induced by excessive levels of calcium. Studies show that overexpression of calcium-permeable ion channels, such as TRPV1, can trigger necrotic cell death in cancer cells [118]. Similarly, compounds such as (−)-englerin A induce cancer cell death by promoting calcium influx, highlighting the potential of calcium channel modulation in cancer therapy [57]. Calcium overload is also important in less common forms of cell death, such as ferroptosis, pyroptosis, and paraptosis. By influencing these pathways, calcium ions offer a multifaceted therapeutic approach to treating cancer and other diseases. Effective manipulation of calcium signaling could therefore enhance treatment strategies by selectively inducing cell death in tumors, reducing inflammation, and preventing cell damage.

We also aim to explore the concept of calcicoptosis as a new form of RCD [7]. Calcium overload, characterized by excessive accumulation of calcium ions within cells, is frequently observed in conjunction with oxidative stress. This oxidative stress is marked by an accumulation of ROS, which further exacerbates cellular damage. The imbalance in calcium homeostasis, often referred to as calcium overload or calcium dysregulation, typically involves an excessive influx or release of calcium ions. While the precise threshold for what constitutes “overload” is not clearly defined, it is understood as a shift from normal cellular calcium balance toward a state that initiates pathological processes. The interaction between calcium overload and oxidative stress is critical in many cell death pathways. This has been particularly evident in studies involving inhibitors of mPTP, such as cyclosporin A and tamoxifen derivatives, which prevent mPTP opening and mitigate calcium-induced damage. These inhibitors, by stabilizing mitochondrial function, not only reduce cell death but also show promise for synergistic effects when combined with therapies targeting oxidative stress or calcium dysregulation. The question arises of whether the term “calcicoptosis” accurately represents a new category of cell death or whether it simply reflects an existing understanding of calcium’s role in cell death pathways. Historically, the involvement of calcium in apoptosis has been described as “calcium-induced apoptosis” or apoptosis via the pseudoreceptor pathway. Mechanisms through which calcium contributes to cell death were identified earlier but were categorized differently. The term “calcicoptosis” introduces a new nomenclature, suggesting that calcium-induced cell death might represent a distinct process rather than a variation of apoptosis or other forms of RCD. In reality, calcium overload acts as a trigger rather than a specific form of cell death in itself. The resultant cell death is highly context-dependent and varies according to the cellular environment and the nature of the stress encountered. Excess calcium influx can enhance the likelihood of initiating cell death, but the cellular context and the interaction with other factors determine the specific type of cell death. Thus, while calcium overload is a significant factor in many forms of RCD, it may not necessarily constitute a new form of cell death per se, but rather represents a mechanism or trigger within existing pathways.

The trend toward identifying new forms of cell death reflects a broader interest in understanding the complexity of cellular responses and the mechanisms underlying various diseases. However, this trend also highlights a need for standardized guidelines to classify and identify cell death mechanisms. The lack of specific criteria for categorizing cell death—whether based on biochemical, functional, or mechanistic factors—creates ambiguity in distinguishing between new and existing forms of cell death. Regulatory bodies like the Committee on Cell Death Nomenclature should develop new guidelines to address these challenges. The development of calcium-based nanoparticles targeting calcium dysregulation and cell death represents a promising and innovative approach in cancer therapy. This strategy leverages the precise control of calcium levels to induce cell death in cancer cells, showcasing its potential effectiveness. Nevertheless, whether referring to this approach as a discovery of a new form of cell death, such as calcicoptosis, adds meaningful scientific value is debatable. Perhaps the focus should remain on the therapeutic potential and efficacy of these novel strategies rather than on the semantic categorization of cell death types.

In summary, while “calcicoptosis” offers an intriguing perspective on calcium’s role in cell death, it is crucial to carefully consider whether it represents a genuinely novel form of cell death or an extension of existing understanding. The importance lies in utilizing these insights to advance therapeutic strategies and improve treatment outcomes.

## 6. Conclusions

Calcium is essential for numerous physiological processes, including muscle contraction, synaptic transmission, and maintaining bone strength. Proper regulation of calcium is critical for health, and its dysregulation can lead to serious diseases, highlighting the need for further research into calcium-related disorders.

Calcium overload is increasingly recognized as a significant contributor to metabolic and proliferative disorders, often leading to oxidative stress and cellular damage. Understanding the interplay between calcium dysregulation and oxidative stress opens new avenues for therapeutic strategies, particularly in cancer treatment, emphasizing the importance of targeted interventions to enhance treatment outcomes.

Calcium overload is a significant factor in various forms of RCD, including apoptosis and necroptosis. Excess calcium release from the ER initiates cellular events such as calpain and caspase activation, which are closely associated with oxidative and ER stress that disrupt cellular homeostasis. Additionally, calcium signaling pathways influence gene expression and cellular stress responses, while excessive calcium influx can trigger necrosis, particularly in cancer cells. Moreover, calcium plays a role in less common forms of cell death, like ferroptosis and pyroptosis, indicating its potential as a therapeutic target. By effectively manipulating calcium signaling, novel treatment strategies could selectively induce cell death in tumors, mitigate inflammation, and protect against cell damage.

The exploration of calcicoptosis as a potential new form of RCD emphasizes the significance of calcium overload and its relationship with oxidative stress. Although excessive calcium accumulation disrupts cellular balance and triggers various cell death pathways, it remains uncertain whether calcicoptosis represents a distinct category of cell death or simply refines existing concepts. Ultimately, calcium overload serves as a trigger rather than a standalone form of cell death, with the specific outcomes highly dependent on cellular context and environmental factors.

The ongoing identification of new forms of cell death reflects a growing interest in understanding cellular responses and disease mechanisms. However, the lack of standardized criteria for categorizing the type of cellular demise creates ambiguity, underscoring the need for clear guidelines. While the development of calcium-based nanoparticles offers innovative cancer therapies by precisely targeting calcium dysregulation, the scientific merit of labeling these strategies as new forms of cell death, like calcicoptosis, remains questionable. The emphasis should be on their therapeutic efficacy rather than on classification.

We hope that future studies will determine whether calcicoptosis represents a distinct form of RCD or a triggering mechanism within existing pathways like apoptosis or necroptosis. To clarify this, rigorous biochemical and molecular criteria are needed, and standardized guidelines from bodies such as the Nomenclature Committee on Cell Death would provide consistency. The precise signaling pathways and molecular players involved in calcicoptosis remain poorly defined, and future research should focus on identifying key regulatory components, such as ion channels, pumps, and signaling molecules. Advanced imaging and single-cell analysis could help uncover how excess calcium interacts with oxidative stress and other cellular phenomena. Understanding the conditions under which calcium overload induces cell death is also crucial, as factors like cell type, metabolic state, and stressors may influence whether apoptosis, necroptosis, ferroptosis, or calcicoptosis occurs. Targeting calcium dysregulation, especially in cancer cells, shows therapeutic potential, with calcium-based nanoparticles and channel modulators offering promising ways to selectively induce cell death in tumors. Besides cancer, calcium dysregulation plays a role in neurodegenerative diseases, cardiovascular disorders, and tissue calcification, presenting broader therapeutic opportunities. Investigating the interaction between calcium signaling and other pathways, such as inflammation or metabolic dysfunction, may also uncover new intervention points. Finally, the scientific community should critically assess whether the term “calcicoptosis” adds value or simply complicates the understanding of RCD, underscoring the need for both basic and applied research to fully unravel calcium’s role in cell death and its potential to transform therapeutic strategies.

## Figures and Tables

**Figure 1 ijms-25-13727-f001:**
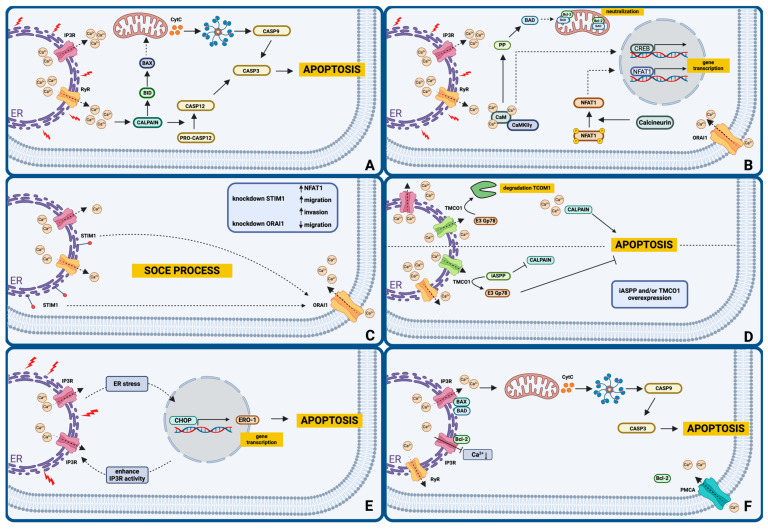
Calcium signaling mechanisms leading to apoptosis. (**A**) Calcium is released into the cytoplasm through inositol triphosphate receptors (IP3Rs) and ryanodine receptors (RyRs) present in the endoplasmic reticulum (ER) membrane in response to ER stress. Free calcium ions interact with calpain, leading to the activation of caspase 12 and subsequently caspase 3, resulting in apoptosis through an ER-related pathway. Calpain also triggers mitochondrial apoptosis by affecting pro-apoptotic proteins Bcl-2 associated X protein (BAX) and BH3 interacting domain death agonist (BID), leading to the release of cytochrome C and the activation of the initiator caspase 9. (**B**) Calcium bound to calmodulin modulates the action of protein phosphatase (PP), which, by influencing the pro-apoptotic protein Bcl2-associated agonist of cell death (BAD), leads to the neutralization of the action of the anti-apoptotic protein BCL-2. Calmodulin is transported to the nucleus by the γ subunit of calcium/calmodulin-stimulated protein kinase II (CaMKIIγ) to phosphorylate transcription factors, including the cAMP-responsive element binding protein (CREB). Furthermore, calcium influx caused by the opening of calcium release-activated calcium channel protein 1 (ORAI1) causes dephosphorylation of nuclear factor of activated T cells 1 (NFAT1) by calcineurin and subsequently leads to its translocation to the nucleus, where it mediates gene expression. (**C**) In the ER membrane, the stromal interaction molecule 1 (STIM1) protein initiates the opening of ORAI1 channels in the plasma membrane, regulates calcium influx, and participates in the store-operated calcium entry (SOCE) process. (**D**) Regulation of calcium homeostasis through the inhibitor of apoptosis-stimulating protein P53 (iASPP)–transmembrane and Coiled-Coil 1 (TMCO1) axis: iASPP competes with the E3 ligase Gp78 to binding to TMCO1, preventing cell apoptosis. (**E**) ER stress affects the function of the C/EBP homology protein (CHOP) pathway, resulting in the expression of ER oxidase 1 alpha (ERO1-α), which increases the activity of IP3R calcium channels, increasing the release of calcium into the cytoplasm. (**F**) The binding of the anti-apoptotic protein BCL-2 to IP3Rs prevents the release of calcium into the cytoplasm. However, the pro-apoptotic proteins BAX and BAD, by acting on calcium channels, promote calcium release, which, in excess, leads to cell death by apoptosis via the mitochondrial pathway. BCL-2 may also interact with other calcium channels, including plasma membrane calcium ATPases (PMCAs). Created with BioRender.com (accessed on 24 October 2024).

**Figure 2 ijms-25-13727-f002:**
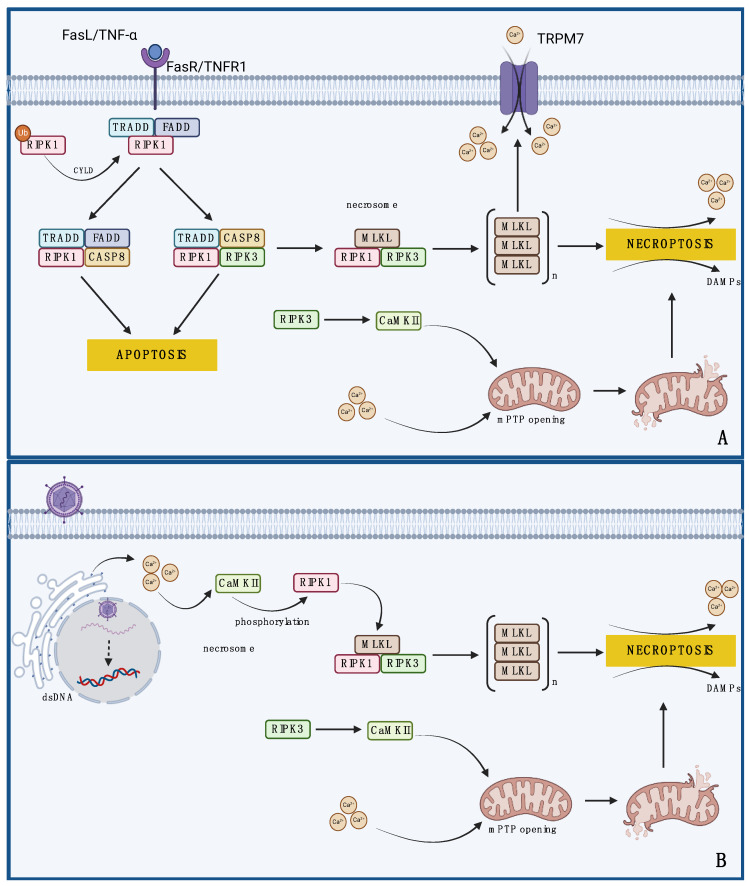
(**A**) Death-receptor-mediated necroptosis: Binding of Fas ligand (FasL) or tumor necrosis factor α (TNF-α) to Fas receptor (FasR) or tumor necrosis factor receptor 1 (TNFR1) causes the assembly of the TNFR1-associated death domain protein (TRADD)/FAS-associated death domain protein (FADD)/receptor-interacting protein kinase (RIPK)-1 complex, where RIPK1 is deubiquitinated by cylindromatosis (CYLD). In the presence of caspase 8 (CASP8), the complex induces apoptotic death. In the absence of CASP8, the complex transforms into a necrosome composed of RIPK1, RIPK3, and lineage kinase domain-like protein (MLKL), leading to necroptosis. During necroptosis, damage-associated molecular patterns (DAMPs) and calcium ions are released, further driving necroptosis. MLKL trimers are translocated to the plasma membrane, where they interact with the transient receptor potential melastatin 7 (TRPM7) channel to promote calcium influx into cells. Excess calcium ions and activated calcium/calmodulin-stimulated protein kinase II (CaMKII) affect the opening of permeability transition pores (PTPs) in mitochondria, leading to their functional and structural degradation and driving necroptosis. (**B**) Death-receptor-independent necroptosis: As a result of the action of factors other than death receptors, e.g., viral infection, there is an increase in the concentration of cytosolic calcium, which activates CaMKII. Kinase phosphorylates RIPK1, leading to necrosome formation and cell death. In addition, the influence of calcium ions and CaMKII can also lead to the opening of PTP channels in the mitochondrial membrane and their structural and functional degradation, enhancing necroptosis. Created with BioRender.com (accessed on 24 October 2024).

**Figure 3 ijms-25-13727-f003:**
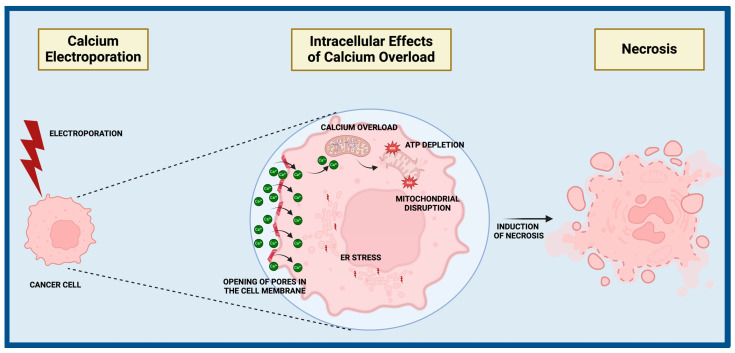
Mechanism and effects of calcium electroporation in cancer therapy. Electroporation increases the permeability of the cell membrane, leading to the opening of pores that allow calcium ions to enter the cell. Normally maintained at a low level inside the cell, calcium ions rapidly increase in concentration. Mitochondria begin to accumulate excess calcium, leading to their overload and subsequent dysfunction. The mitochondrial membrane potential is disrupted, and the mitochondria lose their ability to produce adenosine triphosphate (ATP). Excess calcium also results in the overproduction of reactive oxygen species (ROS), which damage proteins, lipids, and DNA, thereby increasing oxidative stress. The loss of ATP, combined with ROS-induced damage, leads to the destruction of cellular structures and the initiation of necrosis. Created with BioRender.com (accessed on 24 October 2024).

**Table 1 ijms-25-13727-t001:** Calcium-based nanoparticles in targeted cancer therapy.

Type of Nanoparticle	Structure	Mechanism of Action	Type of Induced Cell Death	Ref.
SH-CaO_2_	Calcium peroxide-based nanoparticles coated with sodium hyaluronate (SH-CaO_2_)	Reacts in the acidic tumor environment, releasing calcium ions and hydrogen peroxide, leading to calcium overload and oxidative stress in cancer cells	Apoptosis	[7]
HER-AGIO@CaP-CD	Gelatin-based nanoparticles containing iron oxide, calcium phosphate (CaP), curcumin, and doxorubicin	pH-responsive drug release, dual targeting, synergistic action of curcumin, doxorubicin, and calcium in inducing apoptosis and multidrug resistance	Apoptosis	[142]
M@CaCO_3_@KAE	Calcium carbonate (CaCO_3_)-based nanoparticles with kaempferol-3-O-rutinoside (KAE) and cancer cell membrane coating	Degrades in acidic microenvironment, releasing KAE and calcium ions, disrupting calcium homeostasis, oxidative stress, and mitochondrial destruction	Apoptosis	[143]
Nanoparticles with BAPTA-AM	Nanopolymer with ESCRT inhibitor and calcium ion chelator BAPTA-AM	Chelates calcium ions, prevents cell membrane repair by inhibiting ESCRT III, enhances pyroptosis and cancer cell death	Pyroptosis	[144]
Mito-Jammer	Bimetallic nanoparticle with doxorubicin and calcium peroxide in a metal–organic framework structure	Releases copper and calcium ions, enhances copper-dependent Fenton reaction, mitochondrial dysfunction, overproduction of ROS, intensifies cuproptosis and immune response	Cuproptosis /immunogenic cell death	[141]
Upconversion nanoparticles (UCNPs) coated with ZIF-82	UCNPs coated with a zeolitic imidazolate framework (ZIF-82)	NO release triggered by NIR light, activation of RyRs, Ca^2+^ release from ER, mitochondrial disruption, apoptosis	Apoptosis	[145]
LA-CaO_2_@PDA nanoparticles	Mesoporous calcium peroxide coated with polydopamine and loaded with L-arginine	Calcium peroxide releases H_2_O_2_ that oxidizes L-arginine to produce NO, leading to increased intracellular calcium ions and oxidative stress, while the polydopamine coating enhances photothermal therapy, resulting in immunogenic cell death	Immunogenic cell death	[146]
CaF₂-based luminescent NPs	CaF₂ can be doped with various activators (Yb, Er, or Tm). The doped core is coated with an additional layer, which can be either active or inert	CaF_2_-based upconversion NPs absorb NIR light and re-emit it as visible or UV light, allowing for deep tissue imaging. They label cells for tracking and can encapsulate drugs for controlled release, integrating multiple imaging modalities and therapeutic functions	Apoptosis	[147]
Nano-CaH_2_	CaH_2_ nanoparticles dispersed in polyethylene glycol	Generation of hydrogen, calcium ions, and hydroxide ions through reaction with water; hydrogen therapy, calcium ion overload in cells, neutralization of acidic tumor microenvironment, modulation of tumor microenvironment	Apoptosis	[148]

**Table 2 ijms-25-13727-t002:** Ongoing and future clinical trials investigating calcium electroporation as a therapeutic option for the treatment of cancer. The table provides a clear overview of the study designs, objectives, patient populations, and statuses of the mentioned clinical trial.

Study Identifier	Phase	Objective	Patient Population	Intervention and Evaluation	Endpoints	Status (as of 25 August 2024)
NCT03694080	Phase I	To assess the safety and efficacy of calcium electroporation as a downstaging and immune-response-enhancing treatment before curative surgery.	24 patients with early colorectal cancer (12 rectal, 12 sigmoid) without neoadjuvant chemoradiotherapy indication.	Clinical exams, blood samples, biopsies, and patient-reported outcomes via questionnaires.	Safety, tumor response, immunological changes.	Unknown
NCT03051269	Phase I	To investigate the safety of calcium electroporation in patients with recurrent head and neck cancers.	Patients with recurrent head and neck cancers.	Tumor response via PET/MRI, clinical assessments, biopsies, comparative analysis with electrochemotherapy, and quality of life questionnaires (EORTC QLQ C-30 and H&N35).	Safety, tumor response, quality of life.	Unknown
NCT04259658	Phase II	To explore histopathological mechanisms of tumor cell death and immune responses following calcium electroporation.	24 patients with breast cancer metastases or other cutaneous/subcutaneous malignancies.	TIL population differences in biopsies pre- and post-treatment, immune marker expression, vascular effects, regressive changes, systemic immunological markers.	Differences in TIL populations, immune response analysis.	Completed (results unpublished)
NCT04225767	Phase II	To evaluate the clinical response rate and quality of life impact of calcium electroporation in treating malignant skin tumors.	30 patients with skin metastases and malignant wounds from various cancer histologies.	Single treatment session with calcium electroporation, MR scans, and quality of life interviews.	Overall clinical response rate and quality of life impact.	Active, not recruiting

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
