# Peer review of "The Impact of Calcium Overload on Cellular Processes: Exploring Calcicoptosis and Its Therapeutic Potential in Cancer"

_ijms, 2024, doi:10.3390/ijms252413727_

Round 1
Reviewer 1 Report
Comments and Suggestions for Authors
In this review by Gielecińska, Kciuk, and Kontek, the authors provide a comprehensive analysis of the impact of intracellular calcium dysregulation on cellular processes, emphasizing on its capacity to induce cell death and its potential as an anticancer strategy. The article starts by detailing the role of calcium in various cellular processes, both normal and pathological, and then discuss the emerging field of calcicoptosis. Although the term is relatively recent and challenging to discuss in some aspects, this section is well-structured, approached holistically, and appropriately detailed. Finally, the authors explore some of the most innovative approaches to calcium-based delivery systems for anticancer therapy.
The content is well-organized and logically structured, with high-quality presentation of tables and figures that enhance the manuscript's readability and overall appreciation . I personally enjoyed the manuscript's content and structure; however, I would like to offer a few suggestions that could enrich the current version
- In my opinion, Chapter 2 could omit some of the general pathophysiological aspects of calcium to allow greater emphasis on its role in cancer cells. This omission is particularly important given that the chapter title and the article's focus suggest this should be central. The authors might consider dedicating a brief paragraph to this topic.
- Some additional mechanisms through which compounds can modulate the mitochondrial VDAC channel activity warrant inclusion. For example, compounds that interact directly to promote a high-conductance state facilitate calcium influx and mitochondrial calcium overload, triggering the mitochondrial permeability transition pore opening and subsequent cell death. This is a critical point that should be discussed.
- The role of mitochondrial calcium dysregulation in promoting calcium overload, stress, ROS overproduction, increased autophagy, and cell death should also be highlighted. This has been observed with inhibitors of the mitochondrial permeability transition pore, such as cyclosporin or tamoxifen derivatives. Including these observations would strengthen the discussion. Because mPTP inhbitors may synergize with emergin therapies.
- Emphasizing the vulnerability of cancer cells to mitochondrial calcium overload is particularly important because, unlike all tissues that rely on calcium homeostasis, cancer cells are highly dependent on mitochondrial metabolism (e.g., Warburg effect and metabolic plasticity). This makes them more sensitive to calcium stress than healthy phenotypes.
- Strategies targeting mitochondrial calcium overload can be particularly effective in "hyperdependent" cancer models, such as lymphoid cancers. For instance, lymphocytes, unlike other cells, have a nucleus occupying nearly 80% of the cell volume, with limited cytoplasm and few mitochondria. This unique cellular architecture makes calcium overload especially effective in lymphomas and leukemias. There is evidence supporting this approach, at least in vitro, and it would be valuable to include a brief mention of this evidence in the manuscript.
Reviewer 2 Report
Comments and Suggestions for Authors
The manuscript by Gielecińska et al. entitled “The Impact of Calcium Overload on Cellular Processes: Exploring Calcicoptosis and Its Therapeutic Potential in Cancer” covers the role of calcium in the cell death signaling network. In general, the manuscript is well-written and well-structured. Figures are well-designed and informative. The manuscript deserves to be published. However, some issues should be clarified prior to its acceptance.
Indeed, the concept of “calcicoptosis” is not fully refined in the research community. There are multiple definitions of this event. Some authors define it as a kind of calcium overload stress (PMID: 35849180). At the same time, the others emphasize that calcicoptosis is a special calcium overload-induced tumor cell death (PMID: 38479714). Thus, it can be applied only to tumor cells. Probably, there is no reliable evidence that it can be considered as a distinct RCD. Thus, the idea to consider calciptosis not as a distinct RCD advocated by the authors seems reasonable. It is better to avoid the statement that calciptosis is a novel RCD in the beginning of the abstract and emphasize the controversy of calciptosis definition instead. The same issue concerning the nomenclature should be discussed in Line 56. It should be clearly stated in the Introduction not Discussion that one of the important aims of this review is to figure out whether calciptosis is a distinct RCD or calcium overload is an event observed in multiple RCDs.
Additionally, eryptosis, which is a unique suicidal cell death of erythrocytes, differs from apoptosis of nucleated cells. It does not rely on caspases, and calcium acts as its master regulator (PMID: 38036865). Probably among other RCDs, calcium contribution to eryptosis is the highest. Can eryptosis be considered to be some kind of calcicoptosis? Please provide your thoughts in the section describing the role of calcium in apoptosis.
The section Calcium overload in necrosis should be rephrased. Necrosis is an ACD, which occurs due to mechanistic damage and has no regulatory machinery (PMID: 25236395). Thus, it is suggested to consider the title Calcium overload leads to necrosis.
It is suggested to have a separate subsection describing the crosstalk between different RCDs in relation to calcium ions. Can Ca2+ act as a switch between different RCDs and determine the cell fate?
Were Figures created with BioRender? This should be mentioned in the legends.
Directions for further research in the field should be highlighted.
Minor issues:
- Line 96 and Line 820: Ca2+ - 2+ should be superscripted. Check the subscripts and superscripts in the entire manuscript.
- Caspases are written either with a hyphen or without it, e.g., caspase-3 and caspase 3. Please unify
- Line 596: remove space after /
Reviewer 3 Report
Comments and Suggestions for Authors
This review describes the role of calcium overload in inducing calcicoptosis in cancer cells and its therapeutic potential. The review is well written with minor editorial and grammatical errors throughout the text. Below are some minor issues that should be addressed before the manuscript is published in the IJMS.
1) Pag. 4, 3rd paragraph: calcium release in skeletal muscles occurs via ryanodine receptors located in the sarcoplasmic reticulum. The actual calcium channels (also called dihydropyridine receptors) are found on the membrane of the T-tubules but behave like voltage sensors to activate the release of calcium from the sarcoplasmic reticulum.
2) The text has some minor editorial and grammatical errors (proofreading is recommended). For example:
a) Pages 2, lines 85-87: space after a comma “physiological processes, including regulation of synaptic transmission, muscle contraction, heart rate, blood clotting, structural integrity of bones and teeth, control of hormone secretion, and other metabolic processes in the body.”
b) Pages 2, lines 89: missing space “calcium homeostasis, meticulously_modulating”
Round 2
Reviewer 2 Report
Comments and Suggestions for Authors
- The authors have addressed the comments.